# Paleoceanography and ice sheet variability offshore Wilkes Land, Antarctica – Part 3: Insights from Oligocene–Miocene TEX$_{86}$-based sea surface temperature reconstructions.

**Keywords:**

TEX$_{86}$

Oligocene

Wilkes Land

sea surface temperature

Antarctic ice sheet

Julian D. Hartman[1], Francesca Sangiorgi[1], Ariadna Salabarnada[2], Francien Peterse[1], Alexander J.P. Houben[3], Stefan Schouten[4], Henk Brinkhuis[1,4], Carlota Escutia[2], Peter K. Bijl[1]

[1] Marine Palynology and Paleoceanography, Laboratory of Palaeobotany and Palynology, Department of Earth Sciences, Utrecht University, Princetonlaan 8a, 3584CB Utrecht, The Netherlands

[2] Instituto Andaluz de Ciencias de la Tierra, CSIC/Universidad de Granada, Avenida de las Palmeras 4, 18100 Armilla, Granada, Spain

[3] Applied Geosciences Team, Netherlands Organisation for applied scientific Research (TNO), Princetonlaan 6, 3584CB Utrecht, The Netherlands

[4] NIOZ Royal Netherlands Institute for Sea Research, and Utrecht University, Landsdiep 4, 1797SZ 't Horntje, Texel, The Netherlands

*Correspondence to:* Julian D. Hartman (juulhartman@gmail.com)

**Abstract.** The volume of the Antarctic continental ice sheet(s) varied substantially during the Oligocene and Miocene (~34-5 Ma) from smaller to substantially larger than today, both on million-year and on orbital timescales. However, reproduction through physical modeling of a dynamic response of the ice sheets to climate forcing remains problematic, suggesting the existence of complex feedbacks between cryosphere, the ocean and the atmosphere systems. There is therefore an urgent need to improve the models for better predictions of these systems,

including resulting potential future sea level change. To assess the interactions between cryosphere, ocean and atmosphere, knowledge of ancient sea surface conditions close to the Antarctic margin is essential. Here, we present a new $TEX_{86}$-based sea surface water paleotemperature record measured on Oligocene sediments from Integrated Ocean Drilling Program (IODP) Site U1356, offshore Wilkes Land, East Antarctica. The new data is presented along with previously published Miocene temperatures from the same site. Together the data covers the interval

between ~34 to ~11 Ma and encompasses 2 hiatuses. This record allows us to accurately reconstruct the magnitude of sea surface temperature (SST) variability and trends on both million-year and on glacial-interglacial timescales. On average, $TEX_{86}$ values indicate SSTs ranging between 10 and 21°C during the Oligocene and Miocene, which is on the upper end of the few existing reconstructions from other high-latitude Southern Ocean Sites. SST maxima occur around 30.5, 25 and 17 Ma. Our record suggests generally warm to temperate ocean offshore Wilkes Land.

Based on lithological alternations detected in the sedimentary record, which are assigned to glacial-interglacial deposits, a SST variability of 1.5–3.1°C at glacial-interglacial timescales can be established. This variability is slightly larger than that of deep-sea temperatures recorded in Mg/Ca data. Our reconstructed Oligocene temperature variability has implications for Oligocene ice-volume estimates based on benthic $\delta^{18}O$ records. If the long-term and orbital-scale SST variability at Site U1356 mirrors that of the nearby region of deep-water formation, we argue that

a substantial portion of the variability and trends contained in long-term $\delta^{18}O$ records can be explained by variability in Southern high-latitude temperature and that the Antarctic ice volume may have been less dynamic than previously thought. Importantly, our temperature record suggests that Oligocene-Miocene Antarctic ice sheets were generally of smaller size compared to today.

## 1 Introduction

Numerical paleoclimate models predict that with the current rate of ice volume loss (up to 109±56 Gt/yr, The IMBRIE Team, 2018) several sectors of the West Antarctic marine-based ice sheet will disappear within the coming few centuries (e.g., Joughin et al. 2014; The IMBRIE Team, 2018) favored by ocean warming-induced collapse. Observations show that glaciers on East Antarctica are also vulnerable to basal melt through warming of the ocean waters when they are grounded below sea level (Greenbaum et al., 2015; Miles et al., 2016; Shen et al., 2018; The IMBRIE Team, 2018), making the East Antarctic ice sheet (EAIS) not as stable as previously thought (Mcmillan et al., 2014). Recent numerical modelling studies have improved on reproducing the observed ice-sheet volume decrease, as they incorporate positive feedbacks (e.g., bedrock topography) to global warming and more complicated physics (e.g., hydrofracturing and ice-cliff failure) into these models (Austermann et al., 2015; Deconto and Pollard, 2016; Fogwill et al., 2014; Golledge et al., 2017; Pollard et al., 2015). They indeed show that sensitivity to global warming is particularly high where the ice sheet is grounded below sea level (Fretwell et al., 2013), such as in the Wilkes Land basin (Golledge et al., 2017; Shen et al., 2018).

Both on glacial-interglacial (Parrenin et al., 2013) and longer term Cenozoic timescales (Pagani et al., 2011; Zachos et al., 2008), Antarctic ice-volume changes have been mostly linked to changes in atmospheric $CO_2$ concentrations ($pCO_2$, see e.g., Foster & Rohling 2013; Crampton et al. 2016), modulated by astronomically forced changes in insolation (e.g., Holbourn et al., 2013; Liebrand et al., 2017; Miller et al., 2017; Pälike et al., 2006b; Westerhold et al., 2005). Foster & Rohling (2013) compiled $pCO_2$ proxy data and associated sea level reconstructions for the last 40 million years (Myr). These data suggest that if the past is projected to the future all ice on West Antarctica and Greenland may be lost under current and near future atmospheric $CO_2$ conditions (400-450 ppmv) in equilibrium state. Projections of $pCO_2$ for future emission scenarios of the latest IPCC Report (2014) show a range from 500 to 1000 ppmv for the year 2100, which could lead to additional ice-sheet volume loss from East Antarctica. This range in atmospheric $pCO_2$ is similar to that reconstructed for the warmest intervals of the Oligocene and Miocene epochs (full range: 200–1000 ppmv; e.g., Zhang et al. 2013; Super et al., 2018). Given that observations clearly link the recent instability of marine-based ice sheets to ocean warming, it becomes important to better constrain near-field sea surface temperatures (SSTs) from the Antarctic margin during the Oligocene and Miocene to improve our understanding of past ice sheet dynamics and the projections for the future.

EAIS volume changes have been suggested for the Oligocene and Miocene based on a number of deep-sea $\delta^{18}O$ records, which reflect a combination of bottom-water temperature and ice volume (e.g., Liebrand et al., 2017; Miller et al., 2013; Pekar et al., 2006; Pekar and Christie-Blick, 2008; Shevenell et al., 2004; Westerhold et al., 2005), as well as sedimentary paleosealevel reconstructions (John et al., 2011; Gallagher et al., 2013; Stap et al., 2017). These records show long-term (1-3 Myr) trends punctuated by strong but transient glaciation events (Oi- and Mi-events) (Hauptvogel et al., 2017; Miller et al., 2017; Liebrand et al., 2017, 2016; Pälike et al., 2006b; Westerhold et al., 2005). Following the onset of the Oligocene, marked by the Oi-1 glaciation event, the long-term trend shows a shift towards lighter $\delta^{18}O$ values and a steady increase towards 27 Ma, then a decrease to 24 Ma and a final increase leading into the Miocene, marked by the Mi-1 glaciation event (Beddow et al., 2016; Cramer et al., 2009; Liebrand et al., 2016; Zachos, 2001). Miocene benthic $\delta^{18}O$ long-term trends show a sudden increase at 16.9 Ma, which marks

the onset of the Miocene Climatic Optimum (MCO), a plateau phase, and a subsequent stepwise decrease known as the mid-Miocene Climatic Transition (MCT) (Holbourn et al., 2015, 2013, 2007; Shevenell et al., 2004; Westerhold et al., 2005). The Oligocene and Miocene glaciations are paced by periods of strong 110-kyr eccentricity

fluctuations of up to 1‰ (Liebrand et al., 2017, 2016, 2011). These $\delta^{18}$O fluctuations may be mostly resulting from the waxing and waning of the EAIS, in which case the ice sheet was highly dynamic, or they mostly reflect large changes in deep-sea temperature, in which case large SST fluctuations in the region of deep-water formation were to be expected. Considering the former, fluctuations between 50% and 125% of the present-day EAIS have been suggested for the Oligocene (DeConto et al., 2008; Pekar et al., 2006; Pekar and Christie-Blick, 2008), but this

amount of variability has not yet been entirely reproduced by numerical modeling studies (DeConto et al., 2008; Gasson et al., 2016; Pollard et al., 2015). Considering the latter, several studies have suggested that during the Oligocene the southern high latitudes were the prevalent source for cold deep-water formation (Katz et al. 2011; Goldner et al. 2014; Borelli & Katz 2015). Hence, deep-water temperature records from the southern high-latitudes, particularly those capturing temperature changes on million-year as well as orbital timescales, may provide

information on the relative contribution of deep-sea temperature variability to the $\delta^{18}$O records. However, reconstructions of deep-water temperature based on $\delta^{18}$O and Mg/Ca ratios of benthic foraminifera are hampered by the poor preservation of carbonates on the high-latitude Southern Ocean floor, and rely on critical assumptions on past composition of seawater chemistry. Therefore, one needs to assume that the deep-sea temperature trend captured in the Oligocene and Miocene $\delta^{18}$O records is related to surface water temperature in the Southern Ocean

similarly to today (Baines, 2009; Jacobs, 1991) and in the Eocene (Bijl et al., 2009). Based on this assumption, Southern Ocean SSTs would potentially gauge deep-sea temperature variability. Only few Oligocene SST estimates are available for the Southern Ocean and they relate to the early Oligocene (Petersen and Schrag, 2015; Plancq et al., 2014). Few Southern Ocean SST records are available for the early and mid-Miocene (14–17 Ma) (Kuhnert et al., 2009; Majewski & Bohaty, 2010; Shevenell et al., 2004) and only two (Levy et al., 2016; Sangiorgi et al., 2018) are

derived from south of the Polar Front (PF). Obstacles for reconstructing Oligocene and Miocene SST in the Southern Ocean are the paucity of stratigraphically well-calibrated sedimentary archives, as well as suitable indicator fossils/compounds within these sediments that can be used to reconstruct SST.
        In 2010, the Integrated Ocean Drilling Program (IODP) cored a sedimentary archive at the boundary of the continental rise and the abyssal plain offshore Wilkes Land with a well-dated Oligocene and Miocene sequence:

IODP Site U1356 (Fig. 1), suitable for paleoclimatological analysis. In this study we use the now well-established ratio between several isoprenoidal glycerol dialkyl glycerol tetraethers (GDGTs), the so-called TEX$_{86}$ proxy (Schouten et al., 2013, 2002), to reconstruct SSTs at this high-latitude Southern Ocean site. We present new SST data based on TEX$_{86}$, covering almost the entire Oligocene, along with published TEX$_{86}$ values for the mid-Miocene section (Sangiorgi et al., 2018). Detailed lithological logging of both the Oligocene and Miocene sections of Site

U1356 allows for the distinction of glacial and interglacial deposits (Salabarnada et al., 2018). This enables us to assess long-term evolution of SSTs in proximity of the ice-sheet as well as the temperature differences between glacials and interglacials on orbital time scales, which have implications on the dynamics of the Antarctic ice-sheet and its sensitivity to climate change. We compare our record with the few existing early Oligocene and mid-

Miocene SST data from other high latitude Southern Ocean sites as well as with deep-water $\delta^{18}$O and Mg/Ca-based

bottom water temperature (BWT) records from lower latitudes (Billups & Schrag 2002; Lear et al., 2004; Shevenell et al., 2004), and discuss the implications of our findings.

Together with the companion papers by Salabarnada et al. (2018) and Bijl et al. (2018a) on the lithology and dinocyst assemblages of Site U1356, we contribute significantly to the limited knowledge that exists on Oligocene-Miocene paleoceanographic conditions close to the Antarctic margin.


## 2 Materials & Methods

### 2.1 Site description

Integrated Ocean Drilling Program (IODP) Expedition 318 Site U1356 was cored about 300 kilometers off the Wilkes Land coast (63°54.61'S, 135°59.94'E) at the boundary between the continental rise and the abyssal plain at a

water depth of 3992 m (Escutia et al. 2011, see Fig. 1). Today, this site is south of the Antarctic Polar Front (PF) and is under the influence of by Antarctic Bottom Waters (AABW), Lower Component Deep Water (LCDW), Upper Component Deep Water (UCDW), and Antarctic Surface Water (AASW) (Orsi et al., 1995). Modern-day annual SST values lie around 0°C (summer SSTs are about 1–2°C) (Locarnini et al., 2010).

### 2.2 Sedimentology

At present, IODP Site U1356 receives sediments transported from the shelf and the slope as well as *in situ* pelagic component. Although we have no quantitative constraints on the water depth during the Oligocene and Miocene, the sediments as well as the biota suggest a deep-water setting at Site U1356 during these times (Houben et al., 2013; Escutia et al., 2014). Sedimentary Units of Hole U1356A have been defined in the shipboard report (Escutia et al., 2011). Detailed logging of the sediments recovered in Hole U1356A has revealed that the Oligocene and Miocene

sedimentary record (between 95.40 and 894.80 meters below sea floor, mbsf) consists mostly of alternations of (diatomaceous) laminated and bioturbated sediments, gravity flow deposits, and carbonate beds (Salabarnada et al., 2018; Sangiorgi et al., 2018) (Fig. 2). Gravity flow deposits include Mass Transport Deposits (MTDs) formed by the slump and debris flow sediments of the Miocene, Oligocene and Eocene-Oligocene Transition (EOT), and the late Oligocene-Miocene turbidite type facies as defined by Salabarnada et al. (2018). Samples from the MTDs contain

the largest contribution of reworked older material transported from the continental shelf (Bijl et al., 2018a), while in the other lithologies, this component is reduced or absent.

Between 593.4 and 795.1 mbsf, there are clear alternations between greenish carbonate-poor laminated and grey bioturbated deposits with some carbonate-rich bioturbated intervals. These deposits are interpreted as contourite deposits recording glacial-interglacial environmental variability (Salabarnada et al., 2018). Above 600 mbsf,

sediments mostly consist of MTDs with low to abundant clasts (Fig. 2). However, between the MTDs greenish or grey laminated deposits and greenish or grey bioturbated deposits are preserved. Near the bottom of Unit III as

defined in the shipboard report (around 433 mbsf and below), a different depositional setting is represented with alternations between pelagic clays and (ripple) cross-laminated sandstone beds (Escutia et al., 2011). These sandy (ripple) cross-laminated beds are interpreted as turbidite deposits (Salabarnada et al., 2018). Above these turbidite deposits, there are diatomaceous silty clays that are characterized by an alternation of green laminated and grey homogeneous (bioturbated) silty clays. Apart from their diatom content, these deposits are very similar to the Oligocene alternations between carbonate-poor laminated and carbonate-containing bioturbated deposits, and are therefore interpreted likewise (Salabarnada et al., 2018). Upcore within the Miocene section, the alternations between laminated and homogeneous diatomaceous silty clays become more frequent. In the upper Miocene sections (95.4–110 mbsf) laminations become less clear as the sediments become less consolidated, however green and grey alternations can still be distinguished. The more diatomaceous green deposits are interpreted as interglacial stages. Samples analyzed for $TEX_{86}$ were chosen from all the different lithologies (Fig. 2). In particular the (diatomaceous) laminated and bioturbated deposits were sampled, so we can test whether the glacial-interglacial variability inferred from the lithology is reflected in our $TEX_{86}$ data.

## 2.3 Oligocene and Miocene paleoceanographic setting

The Oligocene and Miocene Southern Ocean paleoceanographic configuration is still obscure and controversial. Some studies suggest that most Southern Ocean surface and deep water masses were already in place by Eocene-Oligocene boundary times (Katz et al., 2011). Neodymium isotopes on opposite sides of Tasmania suggest that an eastward flowing deep-water current was present since 30 Ma (Scher et al., 2015). A westward flowing Antarctic Circumpolar Counter Current (ACCC) was already established during the middle Eocene (49 Ma; Bijl et al. 2013) (Fig. 1). Tasmanian gateway opening also allowed the proto-Leeuwin current (PLC) flowing along southern Australia continue eastward (Carter et al., 2004; Stickley et al., 2004) (Fig. 1). However, numerical modeling studies show that throughflow of the Antarctic Circumpolar Current (ACC) was still limited during the Oligocene (Hill et al., 2013), because Australia and South America were substantially closer to Antarctica (Fig. 1) than today (Markwick 2007). Moreover, tectonic reconstructions and stratigraphy of formations on Tierra del Fuego suggest that following open conditions in the middle and late Eocene, the seaways at Drake Passage underwent uplift starting at 29 Ma and definitive closure around 22 Ma (Lagabrielle et al., 2009). Evidence for active spreading and transgressional deposits in the Tierra del Fuego area record the widening of Drake Passage from 15 Ma onwards. The timing of the Drake Passage opening, which allowed for significant ACC throughflow, is still heavily debated (Lawver & Gahagan 2003; Livermore et al., 2004; Scher & Martin 2006; 2008; Barker et al., 2007; Maldonado et al., 2014; Dalziel 2014). Contourite deposits suggest that strong Antarctic bottom water currents first appeared in the early Miocene (21.3 Ma) and that Weddell Sea Deep Water was able to flow westwards into the Scotia Basin since the middle Miocene (~12.1 Ma) (Maldonado et al., 2003; 2005). It has been suggested that the closure and the reopening of Drake Passage are responsible for the warmer late Oligocene and the Miocene Climatic Optimum (MCO), and the subsequent cooling during the Miocene Climate Transition (MCT), respectively, as inferred from the benthic $\delta^{18}O$ records (Lagabrielle et al., 2009). If the throughflow at the Drake Passage was limited in the late

Oligocene and early Miocene, the Antarctic Circumpolar Counter Current (ACCC) was more dominant than the ACC during these times according to the model study of Hill et al. (2013).

Antarctica was positioned more eastward during the Oligocene and Miocene relative to today (due to true polar wander; van Hinsbergen et al., 2015), and Site U1356 was more to the north during the Oligocene and Miocene compared to today (approximately 59°S at 34 Ma to 61°S at 10 Ma). Reconstructions of the position of the PF based on the distribution of calcareous and siliceous microfossils, place the PF at 60°S during the early Oligocene (Scher et al., 2015), which means that Site U1356 may have crossed the PF between 34 and 10 Ma. The more northerly position of Site U1356 may have facilitated the influence of warmer waters during the mid-Miocene at Site U1356 (Sangiorgi et al., 2018), and therefore bottom-water formation may have been absent or limited at Site U1356. Bottom-water formation is expected in more southerly positioned shallow basins where glaciers extended onto the Antarctic shelf, such as the nearby Ross Sea (Sorlien et al., 2007). However, neodymium isotopes obtained from Site U1356 suggest that bottom water formed offshore the Adélie and Wilkes Land coast during the Early Eocene, which seems in contrast with the globally high temperature of that time (Huck et al., 2017). Modeling studies have, however, suggested that density contrasts created by seasonal changes in SST and salinity (with or without sea ice) may have induced deep-water formation and downwelling around Antarctica (Goldner et al., 2014; Lunt et al., 2010).

## 2.4 Age model U1356

Oligocene sediments were recovered in the section from 894.68 mbsf (first occurrence (FO) *Malvinia escutiana*) to 432.64 mbsf (base of subchron C6Cn.2n) at IODP Hole U1356A (Bijl et al., 2018b). The shipboard age model (Tauxe et al., 2012) was based on biostratigraphy with magnetostratigraphic tie points and chronostratigraphically calibrated to the Geologic Timescale of 2004 (Gradstein et al., 2004). We follow Bijl et al. (2018b), who recalibrated the existing age tie points to the Geologic Timescale of 2012 (GTS2012, Gradstein et al., 2012). The FO of *Malvinia escutiana* (894.68 mbsf; 33.5 Ma; Houben et al., 2011) and the last occurrence (LO) of *Reticulofenestra bisecta* (431.99 mbsf; 22.97 Ma) and the paleomagnetic tie points were used to convert the data to the time domain (see Fig. 4). For the Oligocene-Miocene Boundary, we also follow Bijl et al. (2018b) who infer a hiatus spanning ~22.5-17.0 Ma between Cores 44R and 45R (~421 mbsf). It is unknown whether additional short hiatuses exist within the Oligocene record, but this is likely considering the presence of MTDs (Salabarnada et al., 2018; Fig. S1). In addition, the poor core recovery in some intervals dictates caution in making detailed stratigraphic comparisons with other records.

For the Miocene section of Hole U1356A we follow Sangiorgi et al. (2018), who applied the Constrained Optimization methodology (CONOP) of Crampton et al. (2016) on diatom and radiolarian biostratigraphic events to construct an age model. Based on the application of CONOP to the diatom and radiolarian biostratigraphic events a second hiatus was identified spanning approximately the interval between 13.4 and 11 Ma.

## 2.5 Glycerol dialkyl glycerol tetraether extraction and analysis

In addition to the 29 samples from the Miocene section presented in Sangiorgi et al. (2018), a total of 132 samples from the Oligocene and early Miocene part of the sedimentary record (Table S1) were processed for the analysis of glycerol dialkyl glycerol tetraethers (GDGTs) used for $TEX_{86}$. Spacing varies due to variability in core recovery and GDGT preservation. Furthermore, sampling of disturbed strata was avoided. Sample processing involved manual

powdering of freeze-dried sediments after which lipids were extracted through accelerated solvent extraction (ASE; with dichloromethane (DCM)/methanol (MeOH) mixture, 9:1 v/v, at 100°C and 7.6 x $10^6$ Pa). The lipid extract was separated using $Al_2O_3$ column chromatography and hexane/DCM (9:1, v/v), hexane/DCM (1:1, v/v) and DCM/MeOH (1:1, v/v) for separating apolar, ketone and polar fractions, respectively. Then, 99 ng of $C_{46}$ internal standard was added to the polar fraction, containing the GDGTs, for quantification purposes (cf. Huguet et al.,

2006). The polar fraction of each sample was dried under $N_2$, dissolved in hexane/isopropanol (99:1, v/v) and filtered through a 0.45-μm 4-mm-diameter polytetrafluorethylene filter. After that the dissolved polar fractions were injected and analyzed by high performance liquid chromatography/mass spectrometry (HPLC/MS) at Utrecht University. Most samples were analyzed following HPLC/MS settings in Schouten et al. (2007), while some samples (see Table S1) were analyzed by ultra-high performance liquid chromatography/mass spectrometry

(UHPLC/MS) according to the method described by Hopmans et al., (2016). Only a minor difference between $TEX_{86}$ index values generated by the different methods was recorded by Hopmans et al. (2016) (on average 0.005 $TEX_{86}$ units). Reruns of 5 samples with the new method show an average difference between the two methods of 0.011 $TEX_{86}$ units (see Table S2), which translates to a 0.6°C temperature difference based on $TEX_{86}{}^H$ of Kim et al., (2010) and lies well within the calibration error of 2.5°C. GDGT peaks in the (U)HPLC chromatograms were

integrated using Chemstation software. Sixteen of the 132 samples had too low concentrations of GDGTs to obtain a reliable $TEX_{86}$ value and have been discarded (i.e. with peak height less than 3x background, as well as peak areas below $5*10^3$ mV and $3*10^3$ mV for HPLC-MS and UHPLC-MS, respectively).

We have used the branched and isoprenoid tetraether (BIT) index (Hopmans et al., 2004) to verify the relative contribution of terrestrial GDGTs in our samples, compared to marine GDGTs. As isoprenoid GDGTs (isoGDGTs),

used for the $TEX_{86}$ proxy are also produced in terrestrial soils, albeit in minor amounts, they can alter the marine signal when there is a large contribution of soil organic matter to marine sediments. This contribution can be identified by determining the relative amount of branched GDGTs (brGDGTs), which are primarily soil-derived (Weijers et al., 2006), to that of the isoGDGT crenarchaeol (Hopmans et al., 2004). Samples with BIT index values above 0.3 indicate that the $TEX_{86}$-based temperature may be affected by a contribution of soil-derived isoGDGTs

and thus should be discarded (cf. Weijers et al. 2006), although, a high BIT value can sometimes also result from production of brGDGTs in marine sediments and the water column (Peterse et al., 2009; Sinninghe Damsté, 2016). The composition of the brGDGTs can be used to distinguish between marine and soil-derived GDGT input, in particular by using the #ring$_{tetra}$ index (Sinninghe Damsté, 2016). The #rings$_{tetra}$ index can discriminate between marine and soil-derived brGDGTs as the composition of soil-derived GDGTs typically show high amounts of the

acyclic tetramethylated GDGT-Ia, while a dominance of cyclic tetramethylated (Ib and Ic) brGDGTs has been attributed to *in situ* production within the sediments (Sinninghe Damsté, 2016).

Oxic degradation of GDGTs does not affect the relative amounts of individual isoGDGTs (Huguet et al., 2009; Kim et al., 2009). However, oxic degradation may lead to an increased relative influence of soil-derived isoGDGTs, which could bias the $TEX_{86}$ in different ways depending on the composition of the soil-derived isoGDGTs (Huguet et al., 2009). Higher BIT index values are expected in samples with enhanced amounts of soil-derived isoGDGTs due to oxic degradation, and will be discarded. In addition, we calculated the Methane Index (MI) (Zhang et al., 2011), GDGT-0/crenarchaeol (Blaga et al., 2009; Sinninghe Damste et al., 2009), GDGT-2/crenarchaeol ratios (Weijers et al., 2011), and Ring Index (Zhang et al., 2016) to check for input of methanogenic or methanotrophic archaea, or any other non-temperature related biases to $TEX_{86}$.

## 2.6 $TEX_{86}$ calibrations

The $TEX_{86}$ proxy is based on the distribution of isoGDGTs preserved in sediments (Schouten et al., 2013, 2002). In marine sediments these lipids are assumed to originate from cell membranes of marine Thaumarchaeota, which are one of the dominant prokaryotes in today's ocean and occur throughout the entire water column (e.g. Karner et al. 2001; Church et al. 2010, 2003). Applying $TEX_{86}$ in polar oceans has been challenged by the observation that high scatter in the cold end of the core top dataset for $TEX_{86}$ is present (Ho et al., 2014; Kim et al., 2010). To overcome some of the scatter as well as the non-linearity of the $TEX_{86}$-SST relationship, Kim et al. (2010) proposed two isoGDGT-based proxies and calibrations: $TEX_{86}^L$ and $TEX_{86}^H$. The latter is not considered here as it was particularly developed for low-latitude high-temperature surface waters, and high-latitude core-top values were left out of the calibration (Kim et al., 2010). The former was particularly developed for high-latitude low-temperature surface waters. However, it has been shown that $TEX_{86}^L$ is sensitive to changes in the GDGT-2/GDGT-3 ratio ([2]/[3]), which are unrelated to SST (Taylor et al., 2013; Hernández-Sánchez et al., 2014). Instead these [2]/[3] changes result from changes in the Thaumarchaeotal community structure in the water column, because the community that thrives in deeper (>1000 mbsl) nutrient and ammonia-rich waters, produces significantly more GDGT-2 and thereby introduces a water-depth dependency into the calibration (Taylor et al., 2013; Hernández-Sánchez et al., 2014; Villanueva et al., 2015). Close to the Antarctic margin, the abundance of 'shallow' versus 'deep water' Thaumarchaeotal communities at deep-water sites, like Site U1356 during the Oligocene-Miocene, could be affected by the presence of sea ice and the relative influence of (proto-)UCDW and (proto-)LCDW upwelling. For this reason, also $TEX_{86}^L$-based calibrations are not the focus of our study. Instead, we focus on $TEX_{86}$-based calibrations only. All existing $TEX_{86}^{(H)}$ and $TEX_{86}^L$ calibrations have, however, been applied to our data and are presented as a supplementary figure (Figure S1).

In addition to the $TEX_{86}^L$ and $TEX_{86}^H$ calibrations, Kim et al. (2010) also constructed a linear SST calibration based on $TEX_{86}$ that does include the high-latitude core-top values. Despite the scatter at the cold end of the calibration that results from the inclusion of Arctic surface sediment samples with deviating $TEX_{86}$-SST relations, this calibration (SST = $81.5*TEX_{86} - 26.6$ with a calibration error of $\pm5.2°C$) has been shown to plot onto the annual mean sea surface temperatures of the World Ocean Atlas 2009 (WOA2009; Locarnini et al. 2010) for the surface-sample $TEX_{86}$ values obtained in the Pacific sector of the Southern Ocean (Ho et al., 2014). However, this calibration is likely to be influenced by regional differences in water depth, oceanographic setting and archaeal

communities (Kim et al., 2015; 2016; Tierney and Tingley, 2014; Trommer et al., 2009; Villanueva et al., 2015). In addition, this calibration suffers from regression dilution bias caused by the uncertainty in the measured $TEX_{86}$

values plotted on the x-axis (Tierney & Tingley 2014). Regression dilution bias causes flattening of the slope (Hutcheon et al., 2010) and therefore affects reconstructed $TEX_{86}$-based temperatures at the lower and upper end of the calibration range. Modern-analogue calibration methods exist today to overcome this regression dilution bias as well as some of the regional variability of $TEX_{86}$-SST relationships (Tierney and Tingley, 2015, 2014). These calibrations are based on a Bayesian spatially varying regression model (BAYSPAR), which infers a best estimate

for intersection and slope of the calibration based on an assembly of 20° by 20° spatial grid boxes that statistically fit best with an estimate of the prior distribution of temperature (i.e. the prior) (Tierney and Tingley, 2014). As for deep-time temperature reconstructions this prior cannot be based on modern-day annual mean sea surface temperatures, the BAYSPAR method requires a user-specified mean and variance for this prior (Tierney and Tingley, 2014). The prior for Site U1356 is obtained from recent clumped isotope measurements ($\Delta_{47}$) on planktonic

foraminifers from Maud Rise (ODP Site 689) (Petersen and Schrag, 2015), which show early Oligocene temperatures of 12°C. The BAYSPAR approach (Tierney & Tingley 2014; 2015) selects only those $TEX_{86}$ values from the calibration set of Kim et al. (2010) and an additional 155 core-tops from regional core-top $TEX_{86}$ studies, that are relevant for the study site, thereby generating a more regional calibration. Application of BAYSPAR on Site U1356 using a prior mean of 12°C does, however, result in the exclusion of the high-latitude core-top values. For

this reason we find it useful to compare the BAYSPAR results to the results obtained by using the linear calibration of Kim et al. (2010). The BAYSPAR calibration method provides an estimate for SST and an upper and lower 90% confidence interval. For comparison to the linear calibration of Kim et al. (2010) a standard error (SE) has been calculated from these confidence intervals by assuming a normal distribution around the mean, in which case the 90% confidence interval boundaries can be calculated as the mean plus or minus 1.645 times SE.

Despite these recent efforts in improving the $TEX_{86}$-SST relationship, $TEX_{86}$ is known to overestimate temperatures at high latitudes due to multiple possible biases, such as seasonality (Ho et al., 2014; Schouten et al., 2013) and the incorporation of a subsurface signal (0-200 mbsl) at deep-ocean sites (>1000 mbsl) (Hernández-Sánchez et al, 2014; Huguet et al. 2007; Rodrigo-Gámiz et al., 2015; Yamamoto et al., 2012). There is, however, general consensus that $TEX_{86}$ is able to capture decadal and longer-term temperature trends (Richey and Tierney, 2016), which is why the

main focus of this work is on relative SST changes. Indeed, subsurface export of GDGTs is implicitly incorporated into the global $TEX_{86}$-SST calibration and has therefore no implications for reconstructing SST (Hernández-Sánchez et al., 2014). Highest GDGT fluxes are closely linked to highest organic matter, opal (diatom frustules) and lithogenic particle fluxes (Mollenhauer et al., 2015; Yamamoto et al., 2012) and the lack of production of sinking particles that can incorporate GDGTs formed in deeper waters prevents biasing of surface-sediment $TEX_{86}$ values

towards deep-water temperatures (Basse et al., 2014; Mollenhauer et al., 2015; Yamamoto et al., 2012). Still, particular environmental settings (e.g., upwelling regions, regions with oxygen-depleted deep waters, fresh-water surface waters) might favor the transport of a subsurface temperature (subT) signal to the sediments (Kim et al., 2012a, 2012b; Lopes dos Santos et al., 2010; Mollenhauer et al., 2015). Also for polar oceans it has been suggested that reconstructed temperatures reflect subT, since today Thaumarchaeota are virtually absent in the upper 0-45 m of

Antarctic low-salinity surface waters formed by seasonal retreat of sea ice (Kalanetra et al., 2009). Surface water conditions over Site U1356 during the Oligocene and the MCO were much like present-day regions south of the Subtropical Front (STF) and likely not under the influence of a seasonal sea ice system (see Fig. 1) (Sangiorgi et al., 2018; Bijl et al., 2018a). For these time intervals, there is no reason to believe that surface waters were devoid of Thaumarchaeota due to the presence of sea ice and that $TEX_{86}$ values are influenced by an increased subsurface

signal. For the earliest Oligocene and the MCT, the presence of *Selenopemphix antarctica* suggests that Site U1356 was under the influence of a seasonal sea-ice system (Bijl et al., 2018a). For these periods, a reconstruction of subT values may be more appropriate, but this would still imply that SSTs are warmer. We limit our discussion to the $TEX_{86}$-based reconstructions to SST, notably, because it has been shown that $TEX_{86}$-based SST estimates based on the linear calibration of Kim et al. (2010) obtained from core-top samples from today's sea ice-influenced Southern

Ocean, are in accordance with WOA2009 mean annual SST (Ho et al., 2014). This suggests that the effect of sea ice on surface and subsurface isoGDGT production is incorporated into the linear calibration of Kim et al. (2010). We therefore consider that despite the potential absence of Thaumarchaeota in the surface waters during the early Oligocene and the MCT, the calibration of Kim et al. (2010) does provide a reliable estimate of SST for these time intervals. Moreover, [2]/[3] ratios for the earliest Oligocene and the MCT are relatively low and show much less

variability compared to the rest of the record (Fig. 2), which is opposite to what is expected when the relative influence of deep-water Thaumarchaeota increases (Hernández-Sánchez et al., 2014; Villanueva et al., 2015). Calibrations to subT (Kim et al., 2012a; Kim et al., 2012b; Tierney & Tingley, 2015) are therefore not considered here, but are included in supplementary Figure S1.

     To get an estimate for the long-term average SST trends and confidence levels a Local polynomial regression model

(LOESS) has been applied using R, which is based on the local regression model *cloess* of Cleveland et al. (1992). This method of estimating the long-term average trend is preferred over a running average, because it accounts for the variable sample resolution. For the parameter *span*, which controls the degree of smoothing a value was automatically selected through generalized cross-validation (R-package fANCOVA; Wang 2010).

## 3 Results

<a name="360"></a>360      ### 3.1 Discarding potentially biased $TEX_{86}$ values

     A total of 116 samples spanning the Oligocene and earliest Miocene were analyzed for $TEX_{86}$ in this study. When the Miocene samples of Sangiorgi et al. (2018) are included, the total number of samples with sufficiently high GDGT concentrations is 145. However, only 77 of these 145 $TEX_{86}$ values could be used for SST reconstruction for reasons discussed below.

Although sampling of disturbed strata was avoided, a total number of 46 Oligocene and Miocene samples proved to be obtained from MTDs after detailed logging by Salabarnada et al. (2018). Hence, samples from these beds may not reflect *in situ* material exclusively. For the EOT slumps, this is supported by a high degree of reworked Eocene specimens within the dinoflagellate cyst assemblage below 880.08 mbsf (Houben et al., 2013). In addition, the clast-bearing deposits of Unit II and IV and decimeter-thick granule-rich interbeds of Unit VIII (Fig. 2) are interpreted as

ice-rafted debris (IRD) deposits (Escutia et al., 2011; Sangiorgi et al., 2018), and thus indicate the presence of
icebergs above the site during deposition of these intervals. To avoid potential bias due to allochtonous input and
reworking of older sediments, all samples from MTDs are excluded from the SST reconstructions.
A contribution of terrestrial isoGDGTs can also bias the marine pelagic $TEX_{86}$ signal, and can be verified by the
BIT index (Hopmans et al., 2004; Weijers et al., 2006). In seventeen samples – none of which were derived from

MTDs – the BIT index value was >0.3, which indicates that the reconstructed $TEX_{86}$-temperatures are likely affected
by a contribution of soil-derived isoGDGTs (Weijers et al., 2006; Hopmans et al. 2004). For selected samples, the
composition of brGDGTs was analyzed by UHPLC/MS (Hopmans et al., 2016), which showed that #$rings_{tetra}$ were
below 0.7 (Sinninghe Damsté, 2016) meaning that a significant portion of the brGDGTs was likely soil-derived.
Furthermore, the $TEX_{86}$ signal may be influenced by a potential input of isoGDGTs from methanogenic archaea.

Since methanogenic Euryarchaeota are known to produce GDGT-0 and small amounts of GDGT-1, GDGT-2, and
GDGT-3 (Koga et al., 1998), but not crenarchaeol, such a contribution may be recognized by GDGT-0/crenarchaeol
values >2 (Blaga et al., 2009; Sinninghe Damsté et al., 2009). Similarly, methanotrophic Euryarchaeota may
contribute significant amounts of GDGT-1, GDGT-2 and GDGT-3 that can be identified by values >0.3 for the
Methane Index (MI) (Zhang et al., 2011) and/or GDGT-2/crenarchaeol values >0.4 (Weijers et al., 2011). In total

nineteen non-MTD-derived samples have too high GDGT-0/crenarchaeol ratios, too high Methane Index values, or
too high GDGT-2/crenarchaeol values. Fourteen of these nineteen samples also have too high BIT values, meaning
that in total 22 samples are discarded because of a potential contribution of soil-derived and methanogenic or
methanotrophic archaeal isoGDGT input. As a final exercise, the Ring Index ($|\Delta RI|$) was calculated for our dataset
to identify all other non-temperature related influences on the distribution of isoGDGTs in the samples (Zhang et al.,

2016). In addition to the non-temperature related influences discussed above, these could include oxygen
concentrations (Qin et al., 2015), archaeal growth phase (Elling et al., 2014), ammonia oxidation rates (Hurley et al.,
2016) and ecological factors (Elling et al., 2015). Using $|\Delta RI|>0.6$ as a cutoff, no additional samples were discarded.
All GDGT data and $TEX_{86}$ values, including that of the discarded samples, are presented in the supplementary
material (Figure S2, Table S1).

**3.2 Relation between $TEX_{86}$ values and lithology**

After excluding samples with potentially biased $TEX_{86}$ values, the remaining record shows short-term variability
that is strongly linked to the lithology (see Fig. 2). Sediments from the greenish laminated, carbonate-poor (glacial)
facies produce statistically significant (t-test p-value < 0.005) lower $TEX_{86}$ values than values obtained from the
grey carbonate-rich bioturbated (interglacial) facies. For the entire record, $TEX_{86}$ values are on average 0.50 and

0.53 for the glacial (laminated) and interglacial (bioturbated) lithologies, respectively. Paleoceanographic changes
between glacial and interglacial periods may have affected the community structure of the Thaumarchaeota living
over Site U1356, which may have introduced a non-thermal component to the $TEX_{86}$ record that could contribute to
the observed difference between $TEX_{86}$ values from laminated and bioturbated facies. To test if changes in the
composition of the Thaumarchaeotal community have contributed significantly to the observed difference between

$TEX_{86}$ values from laminated facies and bioturbated facies, a t-test was also performed on the [2]/[3] ratios of

laminated and bioturbated lithologies for the entire record (Fig. 2). No significant difference was observed between the laminated and bioturbated facies (t-test p-value > 0.2).

### 3.3 Oligocene and Miocene long-term sea surface temperature trend

Based on the linear temperature calibration of Kim et al. (2010) (black curve in Fig. 3A), our $TEX_{86}$ values yield
highest temperatures around 30.5 Ma (up to 22.6±5.2°C), 25.5 Ma (up to 25.1°C±5.2°C) and around 17 Ma (up to 19.2±5.2°C), whereas lowest temperatures are recorded between 22 and 23.5 Ma (minimum temperatures are 8.3°C±5.2°C), and around 13 and 10.5 Ma (minima around 7.6±5.2°C). On average SSTs based on the linear calibration of Kim et al. (2010) are 16.6°C, 16.7°C and 10.6°C for the Oligocene, MCO and MCT, respectively. Oligocene SST variability increases significantly (p-value<0.001 in F-test) after 26.5 Ma (see Fig. 3B). Before 26.5
Ma, the variation in the record (2σ) is of 3.6°C, while the 2σ is 6.8°C after 26.5 Ma. We note a strong (9.5°C) SST drop at the lower boundary of what is interpreted to represent subchron C6Cn.2n (23.03 Ma) (see Fig. 2 and 4); at the stratigraphic position of maximum $\delta^{18}O$ values related to Mi-1 in the deep-sea records (Beddow et al., 2016; Liebrand et al., 2011; Pälike et al., 2006b). Unfortunately, due to core recovery issues, limited high-resolution chronobiostratigraphic control in this interval, the age model generally lacks the resolution to identify some of the
other known transient temperature drops in our record (~30 Ma, ~24 Ma) to Oligocene glaciation-related Oi-events (Fig. 4).

The SST record for Site U1356 based on the BAYSPAR model shows the same trend as the SST record generated with the linear calibration, but for SST values below 20.5°C it is offset towards slightly warmer values for SSTs based on the linear calibration. Above 20.5°C, BAYSPAR-based SSTs are slightly offset towards cooler values. On
average, BAYSPAR-based SSTs are 0.8°C warmer than the SSTs based on the linear calibration of Kim et al. (2010), and they have a smaller calibration SE (±4.0°C) (red curve in Fig. 3A). This offset and the smaller calibration error result primarily from the fact that the BAYSPAR-calibration does not take the $TEX_{86}$ values for polar core-tops into account. Instead, it bases its calibration mostly on the modern 30-50° northern and southern latitudinal bands (see map in Fig. 3). Nevertheless, the BAYSPAR-based SSTs lie well within the ±5.2°C SE of the
transfer function from Kim et al. (2010), as well as within the SE of about ±4.0°C for the BAYSPAR SST calibration. Average SSTs for the Oligocene, MCO and MCT based on the BAYSPAR SST calibration are 17.2°C, 17.3°C and 12°C, respectively.

## 4 Discussion

### 4.1 Oligocene and Miocene Southern Ocean sea surface temperatures

Our $TEX_{86}$-derived SST record is the first for the Southern Ocean that covers almost the entire Oligocene. Absolute temperature values are relatively high considering the high-latitude position of Site U1356 (~59°S, van Hinsbergen et al., 2015), but confidence can be obtained from the observation that $TEX_{86}$-based reconstructed SSTs from glacial lithologies are generally lower than those from interglacial lithologies (Fig. 4). Several lines of evidence support the

relatively high Oligocene and mid-Miocene temperatures reconstructed for Site U1356. Dinoflagellate cyst

assemblages from the same site (Sangiorgi et al., 2018; Bijl et al., 2018a) mostly contain taxa related to those found between the Polar Front (PF) and the Subtropical Front (STF) today, where mean annual sea surface temperature is between 8 and 16°C (Prebble et al. 2013). This is on the low end of our reconstructed SSTs for the Oligocene and the MCO, but very comparable to SSTs for the MCT. Furthermore, the abundance of *in situ* pollen of temperate vegetation in these sediments (Sangiorgi et al., 2018; Strother et al., 2017), which are most likely derived from the

Antarctic shores, also suggests a relatively mild climate. Finally, the abundance of pelagic carbonaceous facies in some of the Oligocene interglacial intervals of these high-latitude strata is interpreted to occur under the influence of warmer northern-sourced surface waters at Site U1356 (Salabarnada et al., 2018). Our reconstructed SST values for U1356 add to a picture of globally very warm SSTs during the Oligocene and MCO, as was also reconstructed for the North Atlantic based on $TEX_{86}$ with SST values ranging between 24 and 35°C (Super et al., 2018).

In general, Oligocene SST estimates are higher than the SST estimates reconstructed with other proxies in other high-latitude Southern-Ocean Sites 511 and 689 (Fig. 3A). However, the reconstructed ~12°C (standard error: ±1.1-3.5°C) based on clumped isotopes from Site 689 is derived from thermocline-dwelling foraminifera, whereas the temperature of the surface waters was likely higher than that at the thermocline (Petersen and Schrag, 2015). In addition, when the newest calibration for clumped isotope data is applied (Kelson et al., 2017) also higher

temperature estimates, 12.8-14.5°C, are obtained. Temperature estimates between 6 and 10°C have been obtained from ODP Site 511 (see Fig. 1) based on $U^{K'}_{37}$ (Plancq et al., 2014) and $TEX_{86}$ values (Liu et al., 2009), the latter recalculated with the linear calibration of Kim et al. (2010) and the BAYSPAR calibration used here (Fig. 3A). The influence of the cold Antarctic-derived surface current at Site 511 (Bijl et al., 2011; Douglas et al., 2014) (Fig. 1) might be the reason of these colder estimates. Similar to the Eocene, Site U1356 probably was one of the warmest

regions around Antarctica during the early Oligocene (Pross et al., 2012), situated at a relatively northerly latitude (van Hinsbergen et al., 2015) and still under influence of relatively warm proto-Leeuwin current (PLC, Fig. 1) (Bijl et al. 2011; 2018a).

Sangiorgi et al. (2018) compared $TEX_{86}^{L}$-based reconstructed temperatures (based on the 0-200 depth-integrated calibration of Kim et al., 2012a) from the Miocene section of Site U1356 with Mg/Ca-based SST values from

planktic foraminifera from ODP Site 1171, South Tasman Rise (Shevenell et al., 2004), and $TEX_{86}^{L}$-based seawater temperatures from the ANDRILL (AND)-2A core, Ross Sea (Levy et al., 2016). Based on these temperature reconstructions it was established that temperatures at Site U1356 during the MCO are very comparable to the Mg/Ca-based SSTs from the South Tasman Rise are and a few degrees cooler during the MCT, which was further supported by pollen and dinocyst assemblages (Sangiorgi et al., 2018). We reach the same conclusion based on the

reconstructed SSTs using the linear calibration of Kim et al. (2010) and the BAYSPAR calibration for the Miocene $TEX_{86}$ values from Site U1356 as well as site AND-2A (Fig. 3). Based on these recalibrated SST values, we conclude, like Sangiorgi et al. (2018), that during the MCO there was a much reduced SST gradient between Site U1356 and Site 1171, which were at that time positioned at approximately 60°S and 54°S, respectively (van Hinsbergen et al., 2015). Notably, however, the latitudinal difference between the Site U1356 and Site 1171 after the

MCO increased and may be partly responsible for the increased temperature gradient between the two sites at 14 Ma.

Biota-based temperature reconstructions at such high latitudes are likely skewed towards summer conditions, as has also been suggested for Site 689 and Site 511 (Petersen and Schrag, 2015; Plancq et al., 2014). An important reason for this could be the light limitation at high latitudes during winter (e.g., Spilling et al. 2015), which is unfavorable

for the growth and bloom of phytoplankton, and organisms feeding on phytoplankton. The potential summer bias in high latitude $TEX_{86}$-based SST reconstructions has been discussed extensively for other past warm periods (e.g., Sluijs et al., 2008; Bijl et al., 2009; 2010; 2013). Like in these past warm climates, we expect that primary productivity in the Oligocene Southern Ocean was in sync with seasonal availability of light, irrespective of the presence of sea ice or overall climate conditions. Indeed, isoGDGTs likely require pelleting to sink effectively

through the water column to the ocean floor (e.g., Schouten et al. 2013), and therefore depend on the presence of larger zooplankton that feed on the phytoplankton. As phytoplankton blooms mostly occur during Antarctic summer/autumn, as do their predators (Schnack-Schiel, 2001), we expect the highest isoGDGT fluxes to the sediment during the summer in the Southern Ocean, despite their highest production during a different season (Church et al., 2003; Murray et al., 1998; Richey and Tierney, 2016; Rodrigo-Gámiz et al., 2015). A bias towards

summer temperatures is confirmed by the presence of sea-ice dinoflagellate cysts in some parts of the record, which would suggest SSTs near freezing point during winter.

### 4.2 Long-term Oligocene and Miocene sea surface temperature trends

We aim to explore the implications of the long-term trends in our $TEX_{86}$-based SST record, by placing it in context of the global benthic $\delta^{18}O$ and benthic foraminiferal Mg/Ca-based bottom-water temperature (BWT) records, in

order to infer oceanographic changes or changes in ice volume. Due to the relatively low sample resolution, and discontinuous sampling due to core gaps, in comparison to the complete and quasi-continuous $\delta^{18}O$ records (Beddow et al., 2016; Billups et al. 2004; Hauptvogel et al., 2017; Holbourn et al., 2015; Liebrand et al., 2017, 2016; Pälike et al., 2006a; 2006b), we will here focus on the long-term temperature trends. However, we can use the glacial-interglacial alternations in the lithology, which cover the period between 32 and 10 Ma, to differentiate

between glacial and interglacial SST and assess amplitudes (see Fig. 2). As was mentioned, SSTs derived from the glacial facies show a significantly lower mean than SSTs derived from interglacial facies. Separating glacial and interglacial signals allows us to interpret the long-term SST trend, as this removes a potential sampling bias caused by irregularly spaced glacial or more interglacial samples. To obtain both long-term glacial and interglacial SST trends, LOESS curves are plotted through SST estimates from the glacial and interglacial subsets of the Oligocene

(Fig. 4). For the Miocene we averaged SSTs from glacial and interglacial samples for the three-sample 'cluster' (at ~17, ~13.5 and ~10.5 Ma). For both the SST estimates based on the linear calibration of Kim et al. (2010) and those based on the BAYSPAR calibration, the glacial and interglacial LOESS curves plotted through these SST estimates show the same trend. Therefore, we have chosen to show only the BAYSPAR-based SST data and LOESS curves in Figure 4. The SST LOESS curves based on the linear calibration of Kim et al. (2010) lie slightly below the

BAYSPAR SST LOESS curves (i.e. a 0.9°C offset for the glacial LOESS curve and a 0.4°C offset for the interglacial LOESS curve).

     A global benthic foraminiferal stacked $\delta^{18}O$ curve has been constructed by combining the benthic $\delta^{18}O$ records of the far-field Site 1218, eastern equatorial Pacific (Pälike et al., 2006b); Sites 1264/1265, Walvis Ridge, southeast Atlantic (Liebrand et al., 2017, 2016); Sites 926/929, Ceara Rise, equatorial Atlantic (Pälike et al., 2006a ; Zachos et

al., 2001); Site U1334, eastern equatorial Pacific (Beddow et al., 2016); Site 1090, Agulhas Ridge, Atlantic sector of the Southern Ocean (Billups et al., 2004); Site U1337, eastern equatorial Pacific (Holbourn et al., 2015); and Site 588, southwest Pacific (Flower & Kennett, 1993). To obtain a global benthic $\delta^{18}O$ stack in which the global long-term trends are best represented, we have normalized the data to the Site 1264/1265 record of Liebrand et al. (2017, 2016), on which all records now overlap (Fig.4). Mg/Ca-based BWT records are obtained from Site 1218, eastern

equatorial Pacific (Lear et al., 2004); Site 747, Kerguelen Plateau, Southern Ocean (Billups & Schrag, 2002); and Site 1171, Tasman Rise, Southern Ocean (Shevenell et al., 2004). LOESS curves have been plotted through the benthic $\delta^{18}O$ stack as well as the individual Mg/Ca-based BWT records (Fig. 4).

     The LOESS curves through the glacial and interglacial data show similar trends and show a good resemblance to the global benthic $\delta^{18}O$ stack (Fig. 4), particularly when considering the compromised sample resolution of our record.

The temperature optima and minima in the LOESS curves can be directly linked to periods of relatively low and high benthic $\delta^{18}O$ values (maximum and minimum ice volume/BWT), respectively. Temperatures increase from the earliest Oligocene towards 30.5 Ma, while benthic $\delta^{18}O$ values show a decrease in the same interval, but reach a minimum earlier, around 32 Ma. It is difficult to determine whether SSTs are truly lagging the benthic $\delta^{18}O$ values in this interval or whether this is an artifact caused by the age model in this part of the record. The recorded post-Oi-

1 SST warming coincides with the disappearance of IRD (Escutia et al., 2011) and sea-ice related dinoflagellate cysts (Houben et al., 2013) in the same record (Fig. 4). Following this temperature optimum, there is a cooling trend until a minimum is reached around 28-27 Ma, which coincides with relatively high benthic $\delta^{18}O$ values and the Oi-2a and Oi-2b glacials. Subsequently, there is a warming towards a long-term temperature optimum around 25 Ma, which coincides with a minimum in the benthic $\delta^{18}O$ record known as the late Oligocene warming. This temperature

optimum around 25 Ma is characterized by the influx of the temperate dinocyst species *Nematosphaeropsis labyrinthus* (Bijl et al., 2018a). This seems to indicate a strong influence of northern-sourced surface waters at Site U1356, as this species is currently associated with the Subtropical Front and winter and summer temperatures of 6–13°C and 8–17°C, respectively (Esper and Zonneveld, 2007; Marret and De Vernal, 1997; Prebble et al., 2013).

     Finally, the Oligocene LOESS temperature curves show a cooling towards the Oligocene-Miocene transition at 23

Ma. In comparison to the benthic $\delta^{18}O$ record this cooling trend is rather gradual and starts 1 Myr earlier than the steeper benthic $\delta^{18}O$ record increase that starts at 24 Ma and continuous towards the Mi-1 glaciation. We consider this to represent a realistic climate signal, notably so since the age model is sufficiently well-constrained in this part of the record. Glacial and interglacial averages for the Miocene data clustered around 17, 13.5 and 10.5 Ma show a declining trend and follow the increasing benthic $\delta^{18}O$ trend that characterizes the MCT. High amounts of *N.*

*labyrinthus* within the MCO interval support warm surface water conditions (Sangiorgi et al., 2018). After the MCO, increased amounts of sea-ice dinoflagellates and IRD indicate that Site U1356 came under the influence of

seasonal sea ice, and therefore cooler conditions. However, increases of *N. labyrinthus* after the MCT indicate that warmer northern-sourced waters still periodically influenced Site U1356 (Sangiorgi et al., 2018).

The fact that the LOESS temperature trends mirror the benthic $\delta^{18}O$ record may suggest that (1) changes in the Wilkes Land SST correspond to SST changes in the region of deep-water formation, which is reflected in the benthic $\delta^{18}O$ records, (2) changes in the Wilkes Land SST reflect long-term changes in paleoceanography that simultaneously affect or are related to the size of the Antarctic Ice Sheet (AIS) and therefore the deep-sea $\delta^{18}O$ of the sea water, or a combination of both.

Considering the first, in the modern-day Southern Ocean, bottom water forms through mixing along the Antarctic Slope Front (ASF) of Circumpolar Deep Water (CDW) and High Salinity Shelf Water (HSSW), which forms in consequence of sea-ice formation (Gill, 1973; Jacobs, 1991). Associated with the ASF is the westward flowing Antarctic Slope Current (ASC), which contributes to the bottom-water formation and results from the geostrophic adjustment of Ekman transport to the south, which is driven by the predominantly easterly winds around Antarctica (Gill, 1973). It has been suggested that, after the establishment of its shallower westward flowing counterpart, the Antarctic Circumpolar Countercurrent (ACCC), around 49 Ma (Bijl et al., 2013), an ASC was established near Site U1356 in the early Oligocene (Scher et al., 2015). In areas where sea ice was formed during the Oligocene and Miocene the ASC could have enhanced mixing between HSSW and CDW similarly to today. For the Wilkes Land margin this might have been the case for the earliest Oligocene and MCT where we find sea-ice indicators in the dinoflagellate cyst assemblages (Bijl et al., 2018b; Houben et al., 2013). These sea-ice dinoflagellate cysts seem to indicate that winter temperatures at the Wilkes Land margin were cold enough to allow sea-ice formation and therefore maybe formation of deep waters along the Wilkes Land coast during the earliets Oligocene and MCT. However, most of the record is devoid of sea-ice indicators, suggesting that modern-day process of deep-water formation, is unlikely to have occurred at the Wilkes Land Margin. Still, neodymium isotopes of fossil fish teeth from Site U1356 have suggested that deep-water formation took place at the Adélie and Wilkes Land Margin during the Eocene (Huck et al., 2017), when pollen indicate near-tropical warmth (Pross et al., 2012) . Model studies have suggested that during such warm periods seasonal density differences may still induce deep-water formation or downwelling of waters around Antarctica (Goldner et al., 2014; Lunt et al., 2010). Alternatively and more likely, sea ice may have formed in the cooler Ross Sea and transported along the Wilkes Land coast similarly to today during the Oligocene and Miocene, meaning that deep water formed in the Ross Sea where glaciers extended onto the Antarctic shelf (cf. Sorlien et al., 2007). In that case, the absence of sea-ice dinoflagellate cysts during most of the Oligocene and the MCO at Site U1356 would mean that, in contrast to the earliest Oligocene and MCT, sea ice coming from the Ross Sea was prevented from reaching Site U1356 by too warm winter SSTs. This is in accordance with the relatively warmer (summer) SST values for Site U1356 during most of the Oligocene and the MCO. If the reconstructed SST trends of Site U1356 are representative for the climatic trends of a larger region (i.e. including the Ross Sea as a potential region for deep water formation), this climatic signal may have been relayed to the deep ocean and recorded in the stable oxygen isotope composition of benthic foraminifera in far-field sites. In fact, Southern Ocean-sourced deep waters may have reached as far as the north Pacific during the Oligocene (Borelli & Katz 2015). If this is the case, the consistent long-term trends between the SST of Site U1356 and the benthic $\delta^{18}O$

record would imply that the size of the AIS is less variable on these long-term timescales than the benthic $\delta^{18}$O

record would suggest under the assumption of constant BWT (e.g., Liebrand et al., 2017; see Figure 4): much of the variation will be due to deep-sea temperature variation. The small AIS may have been relatively stable during the Oligocene and Miocene, most likely because there was less marine-based ice in comparison to land-based ice as topographic reconstructions of Antarctica would suggest (Gasson et al., 2016; Wilson et al., 2012).

Only one BWT record is available for the Oligocene, which is based on Mg/Ca ratios from Site 1218 (equatorial

Pacific). Mg/Ca was obtained from the benthic foraminifer *Oridorsalis umbonatus* (Lear et al. 2004; Fig. 4), an infaunal species that is to some extent insulated from long-term changes in carbonate ion concentrations (Ford et al., 2016; Lear et al., 2015). Although absolute temperatures may depend on local factors, such as pore water chemistry, the long-term trends should reflect the trends in BWT (Lear et al., 2015). The BWT record of Site 1218 shows a long-term deep-sea warming between 27 and 25 Ma, similar to our SST record. The temperature optimum at 30.5

Ma in our TEX$_{86}$-based SST record cannot be recognized in the BWT record of Site 1218. Similarly to the benthic $\delta^{18}$O record, an optimum is reached earlier (~32 Ma) and this mismatch could be due to uncertainties in the age model of the lower part of the Oligocene section of Hole U1356A. The continued temperature rise after 25 Ma in the BWT record of Site 1218 is also not observed in our TEX$_{86}$-based temperature trend. This could be because the equatorial Pacific mainly receives bottom water from a warmer Pacific sector of the Southern Ocean, east of the

Tasmanian Gateway, and not from the Wilkes Land margin, and the Pacific sector is influenced by warming. Alternatively, there is an increasing influence of a warmer deep-water mass from elsewhere. Notably, Mg/Ca-based BWTs from Kerguelen Plateau (Site 747) show a temperature optimum around 25 Ma preceding the $\delta^{18}$O minimum at 24 Ma, similar to the SST trend at Site U1356. For the mid-Miocene, BWT records of both Site 747 (Kerguelen Plateau) and Site 1171 (South Tasman Rise) show slowly decreasing trends consistent with decreasing TEX$_{86}$-based

SSTs of Site U1356. The similarities of the three Mg/Ca records to our TEX$_{86}$-based SST record support the transfer of a regional SST signal towards the deep ocean through deep-water formation. However, the temperature differences between temperature optima (e.g., the late Oligocene and MCO) and minima (e.g., the mid-Oligocene and MCT) are much larger for the TEX$_{86}$-based SSTs than for the Mg/Ca-based BWTs. This difference in the degree of change could be explained by the fact that the formation of deep waters during winter is constrained at the lower

end by the freezing point of water, which would limit the degree of change during relatively cold intervals. The degree of change could also be reduced by a shift in the location of deep-water formation to higher latitudes during warmer intervals.

Alternatively, long-term SST trends as well as Southern Ocean BWT trends (Sites 747 and 1171) are governed by large-scale tectonic processes, such as the opening and closure of the Drake Passage, as was suggested by

Lagabrielle et al. (2009). Opening of the Drake Passage could result in increased isolation of the Antarctic continent through the establishment of a (proto-)ACC. In turn, this would result in effective blocking of northerly sourced warmer waters as well as ice sheet expansion thereby resulting in a simultaneous benthic $\delta^{18}$O increase and SST decrease.

As an alternative hypothesis, reconstructed SSTs at Wilkes Land may depend on the volume of the ice sheet in the

hinterland. In that scenario most of the long-term trends in the $\delta^{18}$O record is due to ice volume growth and decline.

A more expanded ice sheet will lower SSTs and enhance the formation of sea ice around Antarctica (Goldner et al.,

2014). Expansion of this cool (proto-)AASW and the ocean frontal systems to lower latitudes during glacials may

have cooled SSTs at Site U1356, while ice-volume decrease and the retreat of the ocean frontal systems during

interglacials may have resulted in warmer SSTs at Site U1356. However, the warmth of even the glacial SSTs in our

SST record, as well as the overall absence of sea-ice indicators during most of the Oligocene in these glacial

intervals, strongly argues against this alternative. Only during the MCT, where dinoflagellate cysts and IRD suggest

an increased influence of icebergs or sea ice (Sangiorgi et al., 2018), the ice sheet may have been large enough

during the glacial periods to allow the influence of cool (proto-)AASW at Site U1356.

**4.3 Sea surface temperature variability at glacial–interglacial time scales**

For the Oligocene, the offset between the glacial and interglacial LOESS curves is constant over time (Fig. 4).

Irrespective of the chosen calibration (i.e. $TEX_{86}$ or BAYSPAR), SSTs are on average 1.5-3.1°C higher during

interglacial intervals than during adjacent glacial times. Notably, this glacial-interglacial SST difference is smaller

than the variability that can be observed in samples from within one sedimentary facies. Particularly around 31 Ma,

where we have a higher sampling resolution within a short interval, temperature variability shows a 5°C range and is

therefore larger than the glacial-interglacial temperature difference. We emphasize that the glacial-interglacial facies

changes reflect obliquity-paced shifts in the position of bottom water currents (Salabarnada et al., 2018).

 This glacial-interglacial SST difference is also smaller than the observed amplitude of the variability in our

temperature record ($2\sigma$ = 3.6°C before 27 Ma), because it takes relatively warm glacials and cool interglacial SST

values into account. Also considering that part of the $2\sigma$ variability is due to the relatively large calibration error of

the BAYSPAR calibration (±4.0°C), the difference of 1.5-3.1°C may be a better representation of average glacial-

interglacial SST variation than the $2\sigma$. It has been shown that glacial-interglacial variability could be overestimated

due to increased GDGT export from deeper waters during Pleistocene glacial periods (Hertzberg et al., 2016). These

authors showed that during the Last Glacial Maximum in the equatorial Pacific a reduced nutrient availability and

primary productivity lowered the nitrite maximum in the water column and consequently the position of highest

Thaumarchaeotal export production. If export of GDGTs consistently took place in deeper waters during the glacial

periods in our record, we would expect that [2]/[3] ratios were higher during the glacials. We show (Fig. 2) that this

is not the case. In addition, dinoflagellate cysts suggest that the surface waters overlying Site U1356 become more

oligotrophic during the interglacial periods, meaning that the nitrite maximum in the water column and therefore

GDGT export production would be deeper during the interglacials. Interglacial $TEX_{86}$-derived temperatures would

hence be underestimating true SST values.

The average consistent offset between glacial and interglacial values seems to disappear for each of the Miocene

data clusters at ~17, ~13.5, and ~10.5 Ma. Several causes could explain this. It could be the result of a less variable

climate during the Miocene, which causes both subsets to overlap more. Indeed, for the MCO this may be the case,

because the MCO is a time interval of exceptional warmth, with retreated ice sheets and vegetated coastlines of

Wilkes Land (Sangiorgi et al., 2018). In such a climate, the glacial intervals may not have been fundamentally

colder than the interglacials. We cannot, however, explain the apparent absence of glacial interglacial-temperature

variability around 14 and 10.5 Ma, where dinoflagellate cysts suggest profound variability in sea-ice extent, upwelling and temperature (Sangiorgi et al., 2018). It could be that the samples taken by Sangiorgi et al. (2018) do not capture the true glacial and interglacial extremes, but this cannot be verified at this stage. Because a detailed

lithological log was not available to Sangiorgi et al. (2018), there is also an uneven distribution between glacial and interglacial samples.

If the recorded glacial-interglacial SST variability in the Oligocene is representative for the SST variability at the region of deep-water formation, it should be considered when interpreting benthic foraminiferal $\delta^{18}O$ records in terms of ice-volume variability. As such, a larger part of the variability of $\delta^{18}O$ than so far assumed (Hauptvogel et

al., 2017; Liebrand et al., 2017;) should be ascribed to deep-sea temperature rather than ice-volume changes. If the region of deep-water formation experienced the same SST variability, 40-70% of the 1‰ deep-sea $\delta^{18}O$ variability over Oligocene glacial-interglacial cycles can be related to deep-sea temperature. However, it is plausible that not the entire range of SST variability is relayed to the deep-sea, and that in the more southerly positioned Ross Sea, the most likely region of Oligocene deep-water formation, temperatures were not as variable as in the Wilkes Land

sector. Indeed, Mg/Ca-based reconstructed bottom-water temperatures from Site 1218 show much less glacial-interglacial variation (1.1°C, Fig. 5) (Lear et al., 2004) than our record. Still, our record provides evidence that polar SST experienced considerable variability, both on the short-term glacial-interglacial cycles as well as on the long-term.

A considerable influence of deep-sea temperature on benthic $\delta^{18}O$ could explain the level of symmetry in glacial-

interglacial cycles in the Oligocene (Liebrand et al., 2017), as the temperature would vary in a sinusoidal fashion, whereas ice sheets would respond non-linearly to climate forcing. The sedimentary record of Site U1356 lacks the potential of obtaining a resolution comparable to those of deep-sea $\delta^{18}O$ records, in order to verify these claims. However, ice-volume reconstructions from $\delta^{18}O$ records on both long-term and short-term time scales should consider that an important component of the signal could potentially be ascribed to temperature variability.

**5 Conclusions**

We reconstruct (summer-biased) SSTs of around 17°C on average for the Wilkes Land Margin during the Oligocene, albeit with a high degree of variability (up to a 6.8°C double standard deviation during the late Oligocene). The reconstructed temperatures are a few degrees higher than previously published high-latitude early Oligocene Southern Ocean estimates. Because alternations in the lithology reflect glacial-interglacial cycles, an

estimated temperature difference of 1.5 to 3.1°C between glacials and interglacials could be interpreted for the Oligocene. The long-term trends of both glacial and interglacial records show a temperature increase towards 30.5 Ma, followed by a minimum around 27 Ma, an optimum around 25 Ma and finally a decrease towards the end of the Oligocene, generally following the long-term trends in the global benthic $\delta^{18}O$ record as well as parts of the available Mg/Ca-based BWT records for the Oligocene. Recalibrated SSTs based on previously published TEX$_{86}$

data for the mid-Miocene decrease from around 17°C to 11°C between ~17 and ~10.5 Ma. A distinct glacial-interglacial SST difference was not observed for the mid-Miocene. Nevertheless, the recorded temperature decline

also follows the trend observed in benthic $\delta^{18}O$ and Mg/Ca-based BWT records. Our results suggest that considerable SST variability prevailed during the Oligocene and Miocene. This may have implications for the dynamics of marine-based ice sheets, if present, and the extent of the Antarctic ice sheet in general. Assuming that

the reconstructed SST trends and glacial-interglacial variability have been relayed to the deep water at nearby bottom-water formation sites, our results indicate that the long-term $\delta^{18}O$ trend may be controlled for a considerable part by bottom-water temperature. This implies that the Antarctic Ice Sheet was less dynamic during the Oligocene and Miocene, which could be due to the presence of relatively more land-based versus marine-based ice.

## 6 Acknowledgements

JDH, FS, HB and PKB acknowledge NWO Netherlands Polar Program project number 866.10.110. SS was supported by the Netherlands Earth System Science Centre (NESSC), funded by the Dutch Ministry of Education, Culture and Science (OCW). PKB and FP received funding through NWO-ALW VENI grant no 863.13.002 and 863.13.016, respectively. CE and AS thank the Spanish Ministerio de Econimía y Competitividad for Grant CTM2014-60451-C2-1-P. We thank Alexander Ebbing and Anja Bruls for GDGT sample preparation during their

MSc research. We thank Stephen Gallagher and two anonymous reviewers for their thorough review and constructive comments, which helped improving this manuscript.

## 7 Author contributions

FS, PKB, HB and SS designed the research. JDH, PKB and AJPH carried out Oligocene GDGT analyses. CE and AS incorporated the lithological data. PKB, FP and SS assisted in GDGT analytical procedures and interpretation.

JDH wrote the paper with input from all authors.

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

Limitations ( N , P and Si ) on Growth , Stoichiometry and Photosynthetic Parameters of the Cold-Water

Diatom Chaetoceros wighamii. PLoS One 10, e0126308. https://doi.org/10.1371/journal.pone.0126308

Stap, L.B., van de Wal, R.S.W., de Boer, B., Bintanja, R., Lourens, L.J., 2017. The influence of ice sheets on

temperature during the past 38 million years inferred from a one-dimensional ice sheet-climate model.

Clim. Past 13, 1243–1257.

Stickley, C.E., Brinkhuis, H., Schellenberg, S.A., Sluijs, A., Röhl, U., Fuller, M., Grauert, M., Huber, M., Warnaar,

J., Williams, G.L., 2004. Timing and nature of the deepening of the Tasmanian Gateway.

Paleoceanography 19, 1–18. https://doi.org/10.1029/2004PA001022

Strother, S.L., Salzmann, U., Sangiorgi, F., Bijl, P.K., Pross, J., Escutia, C., 2017. A new quantitative approach to

identify reworking in Eocene to Miocene pollen records from offshore Antarctica using red fluorescence

and digital imaging. Biogeosciences 14, 2089–2100. https://doi.org/10.5194/bg-14-2089-2017

Super, J.R., Thomas, E., Pagani, M., Huber, M., O'Brien, C., Hull, P.M., 2018. North Atlantic temperature and

$pCO_2$ coupling in the early-middle Miocene. Geology 46(6), 519–522.

Tauxe, L., Stickley, C.E., Sugisaki, S., Bijl, P.K., Bohaty, S.M., Brinkhuis, H., Escutia, C., Flores, J.A., Houben,

A.J.P., Iwai, M., Jiménez-Espejo, F., McKay, R., Passchier, S., Pross, J., Riesselman, C.R., Röhl, U.,

Sangiorgi, F., Welsh, K., Klaus, A., Fehr, A., Bendle, J.A.P., Dunbar, R., Gonzàlez, J., Hayden, T.,

Katsuki, K., Olney, M.P., Pekar, S.F., Shrivastava, P.K., van de Flierdt, T., Williams, T., Yamane, M.,

2012. Chronostratigraphic framework for the IODP Expedition 318 cores from the Wilkes Land Margin:

Constraints for paleoceanographic reconstruction. Paleoceanography 27, PA2214.

https://doi.org/10.1029/2012PA002308

Taylor, K.W.R., Huber, M., Hollis, C.J., Hernandez-Sanchez, M.T., Pancost, R.D., 2013. Re-evaluating modern and

Palaeogene GDGT distributions: Implications for SST reconstructions. Glob. Planet. Change 108, 158–174.

https://doi.org/10.1016/j.gloplacha.2013.06.011

Tierney, J.E., Tingley, M.P., 2015. A TEX 86 surface sediment database and extended Bayesian calibration 1–10.

https://doi.org/10.1038/sdata.2015.29

Tierney, J.E., Tingley, M.P., 2014. A Bayesian, spatially-varying calibration model for the TEX86 proxy. Geochim.

Cosmochim. Acta 127, 83–106. https://doi.org/10.1016/j.gca.2013.11.026

Tierney, J.E., Sinninghe Damsté, J.S., Pancost, R.D., Sluijs, A., Zachos, J.C., 2017. Eocene temperature gradients.

Nat. Geosci. 10, 538–539.

The IMBRIE Team, 2018. Mass balance of the Antarctic Ice Sheet from 1992 to 2017. Nature 558, 219–222.

Trommer, G., Siccha, M., van der Meer, M.T.J., Schouten, S., Sinninghe Damsté, J.S., Schulz, H., Hemleben, C.,

Kucera, M., 2009. Organic Geochemistry Distribution of Crenarchaeota tetraether membrane lipids in

surface sediments from the Red Sea. Org. Geochem. 40, 724–731. https://doi.org/10.1016/j.orggeochem.2009.03.001

Villanueva, L., Schouten, S., Damsté, J.S.S., Box, P.O., 2015. Depth-related distribution of a key gene of the tetraether lipid biosynthetic pathway in marine Thaumarchaeota 17, 3527–3539. https://doi.org/10.1111/1462-2920.12508

Wang, Xiao-Feng, 2010. fANCOVA: Non-parametric analysis of covariance. https://CRAN.R-project.org/package=fANCOVA

Warnaar, J., 2006. Climatological implications of Australian-Antarctic separation [Ph.D. Thesis], Utrecht University, pp.143.

Weijers, J.W.H., Lim, K.L.H., Aquilina, A., Sinninghe Damsté, J.S., Pancost, R.D., 2011. Biogeochemical controls
on glycerol dialkyl glycerol tetraether lipid distributions in sediments characterized by diffusive methane flux. Geochemistry, Geophys. Geosystems 12, Q10010. https://doi.org/10.1029/2011GC003724

Weijers, J.W.H., Schouten, S., Hopmans, E.C., Geenevasen, J.A.J., David, O.R.P., Coleman, J.M., Pancost, R.D., Sinninghe Damsté, J.S., 2006. Membrane lipids of mesophilic anaerobic bacteria thriving in peats have typical archaeal traits. Environ. Microbiol. 8, 648–657. https://doi.org/10.1111/j.1462-2920.2005.00941.x

Westerhold, T., Bickert, T., Röhl, U., 2005. Middle to late Miocene oxygen isotope stratigraphy of ODP site 1085 (SE Atlantic): new constraints on Miocene climate variability and sea-level fluctuations. Palaeogeogr. Palaeoclimatol. Palaeoecol. 217, 205–222.

Wilson, D.S., Jamieson, S.S.R., Barrett, P.J., Leitchenkov, G., Gohl, K., Larter, R.D., 2012. Antarctic topography at the Eocene–Oligocene boundary. Palaeogeogr. Palaeoclimatol. Palaeoecol. 335–336, 24–34.
https://doi.org/10.1016/j.palaeo.2011.05.028

Yamamoto, M., Shimamoto, A., Fukuhara, T., Tanaka, Y., Ishizaka, J., 2012. Glycerol dialkyl glycerol tetraethers and TEX86 index in sinking particles in the western North Pacific. Org. Geochem. 53, 52–62. https://doi.org/10.1016/j.orggeochem.2012.04.010

Zachos, J., 2001. Trends, Rhythms, and Aberrations in Global Climate 65 Ma to Present. Science (80-. ). 292, 686–
693. https://doi.org/10.1126/science.1059412

Zachos, J., Shackleton, N.J., Revenaugh, J.S., Pälike, H., Flower, B.P., 2001. Forcing across the Oligocene-Miocene Boundary. Science 292, 274–278.

Zachos, J.C., Dickens, G.R., Zeebe, R.E., 2008. An early Cenozoic perspective on greenhouse warming and carbon-cycle dynamics. Nature 451, 279–283. https://doi.org/10.1038/nature06588

Zhang, Y.G., Pagani, M., Liu, Z., Bohaty, S.M., DeConto, R., 2013. A 40-million-year history of atmospheric CO2. Phil. Trans. R. Soc. A 371, 20130096. https://doi.org/10.1098/rsta.2013.0096

Zhang, Y.G., Pagani, M., Wang, Z., 2016. Ring Index: A new strategy to evaluate the integrity of TEX86 paleothermometry. Paleoceanography 31, 220–232. https://doi.org/10.1002/2015PA002848.Received

Zhang, Y.G., Zhang, C.L., Liu, X., Li, L., Hinrichs, K., Noakes, J.E., 2011. Methane Index : A tetraether archaeal
lipid biomarker indicator for detecting the instability of marine gas hydrates 307, 525–534. https://doi.org/10.1016/j.epsl.2011.05.031

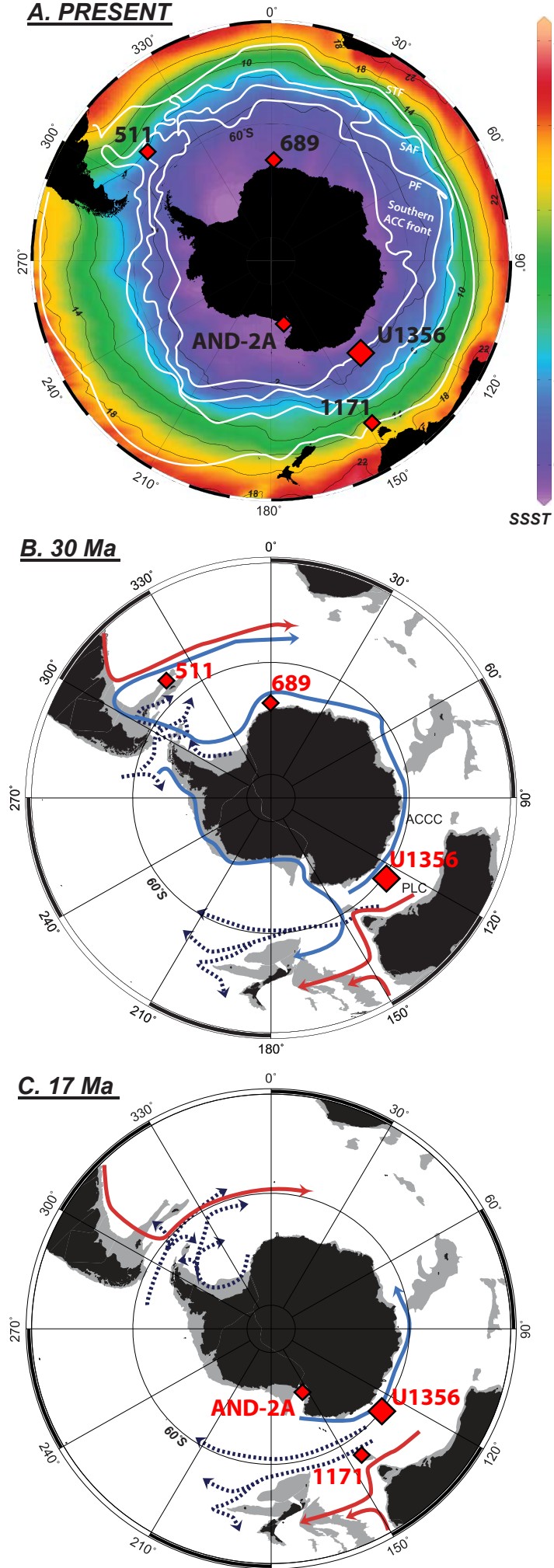

**A. PRESENT**

**B. 30 Ma**

**C. 17 Ma**

**Figure 1:** (A) Present-day Southern Ocean summer temperatures and geography obtained from the World Ocean Atlas (Locarnini et al., 2010) using Ocean Data View, and Southern Ocean Fronts obtained from Orsi et al. (1995). PF = Polar Front, SAF = Sub-Antarctic Front, STF = Sub-Tropical Front. Red diamonds indicate DSDP/ODP/IODP Site locations. (B) Modified ODSN-generated map of Antarctica around 30 Ma (continents in black, shelf areas in grey). Paleolatitudes calculated with paleolatitude.org Hinsbergen et al., 2015). Reconstructed cold (light blue) and warm (red) surface currents are based on publications by Stickley et al. (2004), Warnaar (2006), Bijl et al. (2011), Bijl et al., (2013), and Douglas et al. (2014). Reconstructed bottom-water currents (dotted dark blue) are based on publications by Carter et al. (2004), Livermore et al. (2007), Maldonado et al. (2014) and Scher et al. (2015). ACCC = Antarctic Circumpolar Counter Current, PLC = proto-Leeuwin Current. (C) The same as the middle figure, but then for the period around 17 Ma.

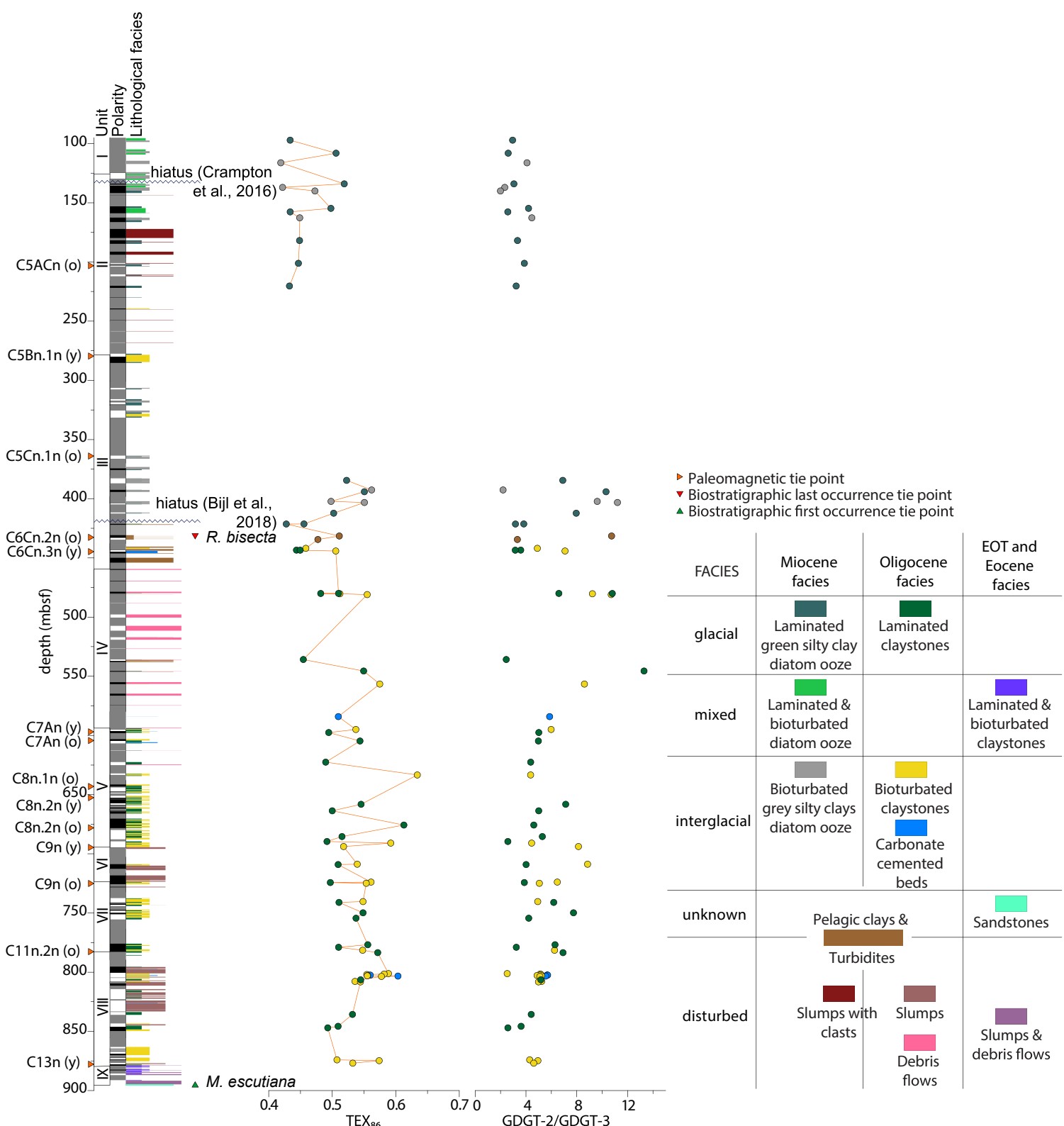

**Figure 2:** Lithology, TEX$_{86}$ values and GDGT-2/GDGT-3 ratios of Hole U1356A plotted against depth (mbsf) with units according to Escutia et al., 2011 and chronostratigraphic tie points and paleomagnetic polarities obtained from Tauxe et al. (2012), adjusted by Crampton et al. (2016) and Bijl et al. (2018). Depositional facies and interpretation are indicated with colors following Salabarnada et al. (2018), see legend to the right. Colors of the TEX$_{86}$ and GDGT-2/GDGT-3 values reflect the lithology they have been sampled from.

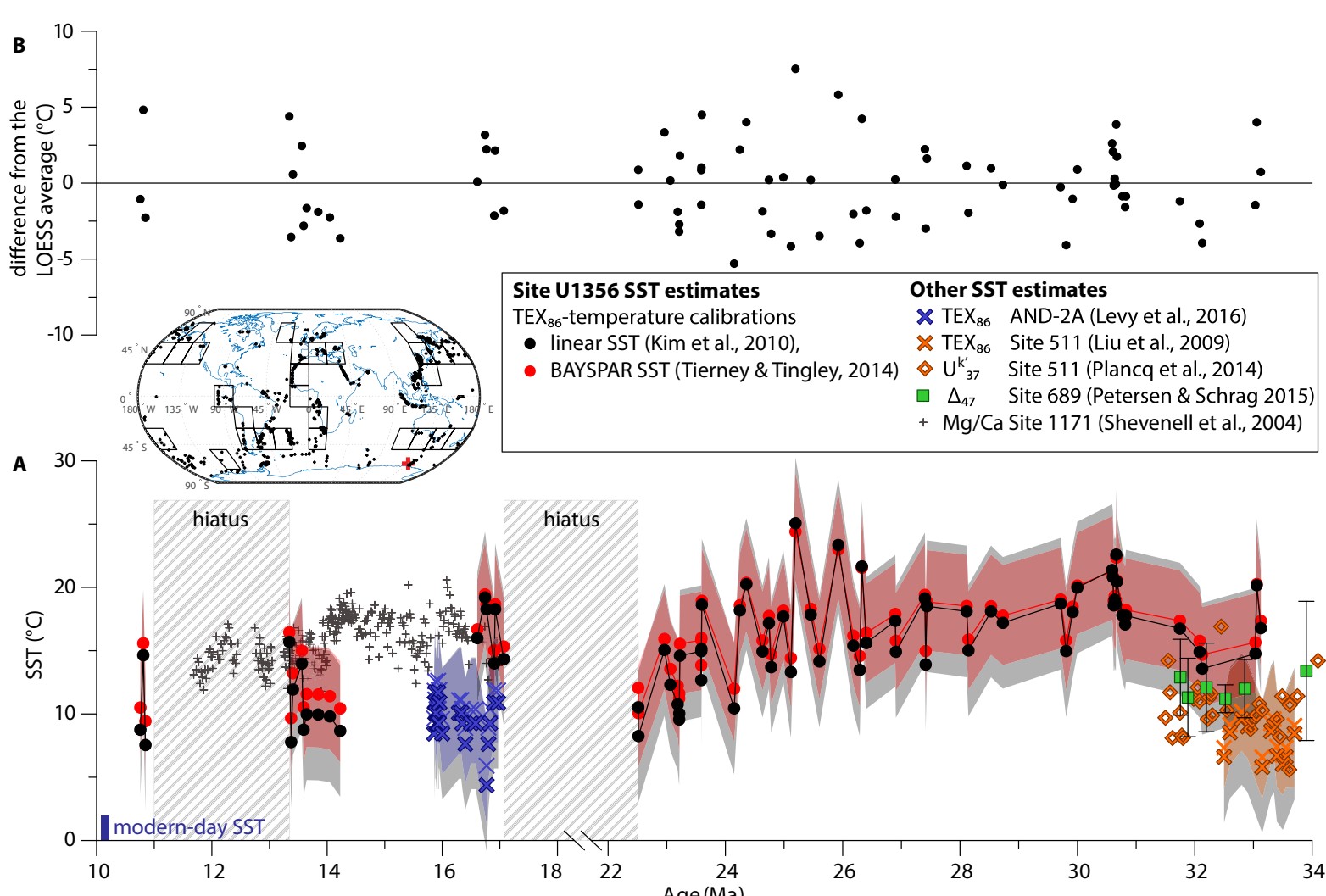

**Figure 3:** (A) TEX$_{86}$-based SSTs plotted against age (Ma), using the linear calibration of Kim et al. (2010) and the BAYSPAR method (see text). The map (center left) indicates the 20°x20° grid cells from which core-top values (black dots) were obtained for the BAYSPAR calibration. The red + in the map indicates the location of Site U1356. Other SST estimates are plotted for comparison. For the TEX$_{86}$-based temperature estimates (blue and orange crosses), the symbols with both fill and outline are based on the linear calibration of Kim et al. (2010), those with the fill color only are based on the BAYSPAR SST calibration (Tierney & Tingley, 2014). The standard error of the Kim et al. (2010) calibration is indicated by grey shading and the standard error of the BAYSPAR calibration is indicated by coloured shading. (B) The SST difference from the LOESS average based on the linear calibration of Kim et al. (2010) plotted against age. Modern-day temperature range indicated along the y-axis.

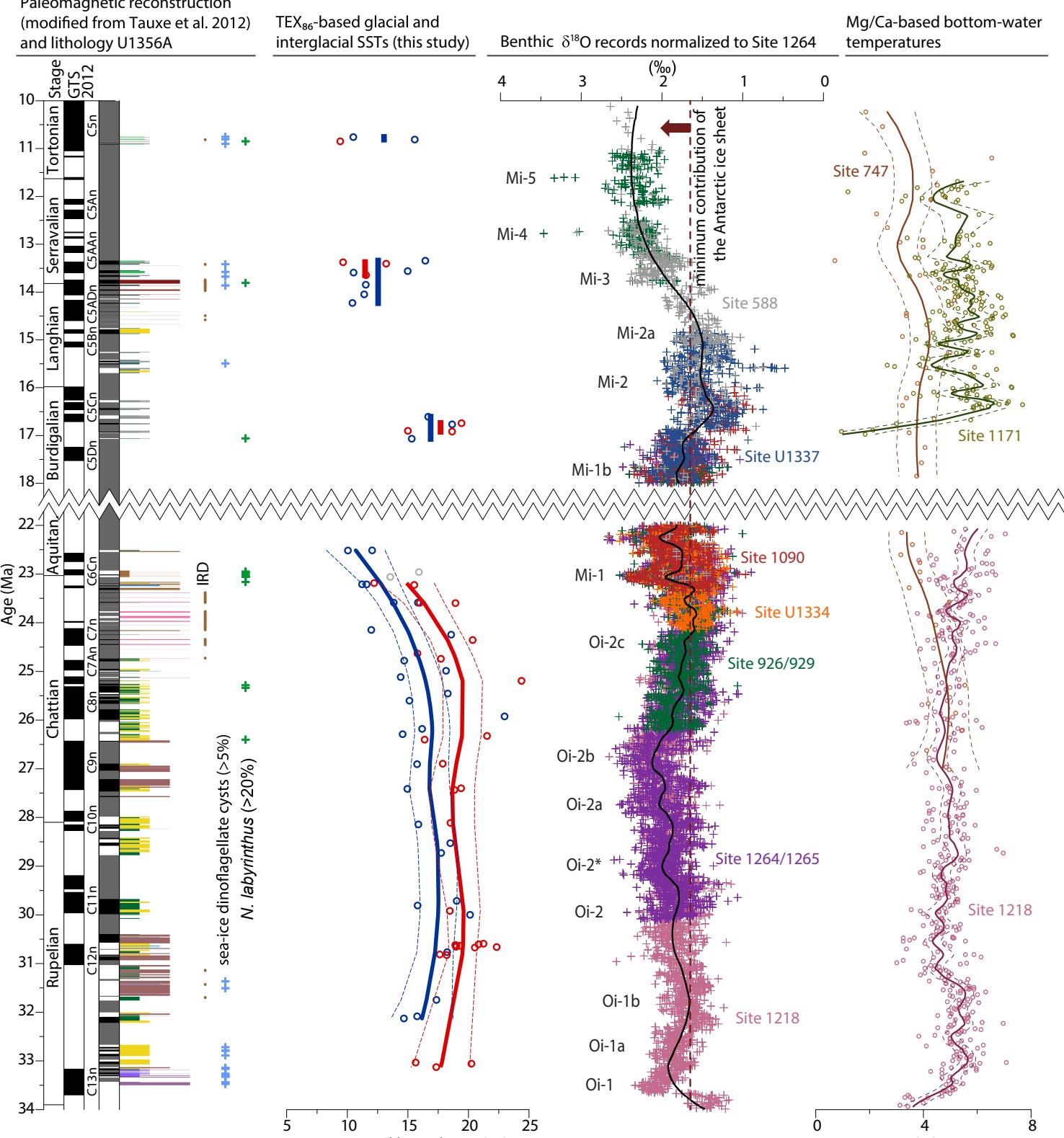

**Figure 4:** Lithology and paleomagnetic polarities as in Fig. 2, but plotted against age (Ma), updated to the GTS2012 timescale (Gradstein et al., 2012). Sections with IRD, and samples with sea-ice and *Nematosphaeropsis labyrinthus* dinoflagellate cysts are indicated. SST values are based on the BAYSPAR calibration with blue and red open circles indicating glacial and interglacial temperature estimates, respectively, based on the lithology (see text). For the Oligocene, thick blue and red lines are the LOESS average glacial and interglacial trends with their 95%-confidence interval (dotted lines). For the Miocene thick blue and red straight lines indicate glacial and interglacial means for the corresponding data clusters. Benthic δ¹⁸O records are normalized to the δ¹⁸O record of Site 1264 (in purple). Colored +-signs indicate the different DSDP/ODP/IODP Sites (refs. in text) and the black line is the LOESS average. Oi- and Mi-glaciation events are indicated. The area below the benthic δ¹⁸O record on the left side of the dark red dotted line (1.65‰) indicates the minimum contribution (in ‰) of the Antarctic Ice Sheet to the benthic δ¹⁸O record, assuming BTW at Site 1264 did not drop below its modern-day value (see Liebrand et al. (2017)). Mg/Ca-based BWT records of three ODP/IODP Sites with LOESS averages (solid lines) and 95% confidence intervals (dotted lines) on the far right (refs. in text).