# Peer review of "Paleoceanography and ice sheet variability offshore Wilkes Land, Antarctica – Part 3: Insights from Oligocene–Miocene TEX86-based sea surface temperature reconstructions."

_Climate of the Past, 2017_

## Referee Comment (RC1) · Anonymous Referee #1 · 24 Jan 2018

In this paper, Hartman et al. use the TEX86 proxy to reconstruct sea surface temperatures (SST) along the East Antarctica margin during the Oligocene epoch. TEX86-derived SST estimates range from 10 to 21°C and are broadly consistent with the dinocyst assemblage (Bijl et al., in review; this volume). However, TEX86 values can fluctuate significantly (up to 5-8°C) over relatively short time intervals (i.e. < 500kyr), suggesting additional controls on the GDGT distributions. Part of this variability could be related to glacial/interglacial climatic variability (Salabarnada et al; in review); however, it cannot explain all of the variance. The authors then compare TEX86-derived SST estimates to published deep-sea delta18O estimates and argue that a significant portion of the deep-sea delta18O record can be attributed to temperature change. This implies that variations in Antarctica ice volume may have been less severe than previously thought.

The development of a long-term Oligocene SST record is a welcome contribution to the Cenozoic paleoclimate community. However, the TEX86 is quite complex and highly variable, suggesting additional controls on the GDGT distributions. The paper is also quite speculative, especially when discussing the potential interplay between temperature and ice volume (e.g. lines 421-439). Understanding the link between temperature and ice volume is a very important question; however, I do not think that this low-resolution (and complex) dataset can answer this question satisfactorily.

Finally, several key conclusions (e.g. interglacial/glacial cyclicity) rely upon two companion papers which are currently in review (Salabarnada et al; Bijl et al). Until these are published, I unable to fully assess the veracity of this paper.

I have explored some of these comments in more detail below:

1) Temperature and ice volume

In Section 4.2, the authors attempt to use TEX86 SST estimates to disentangle temperature and ice volume from the deep-sea delta18O signal. However, there are several caveats to this approach. These include: (1) a possible summer bias in TEX86 estimates, (2) uncertainties in the location of deep-water formation, (3) a "poor age model" (the authors words, not mine!) and (4) low sampling resolution (line 416 to 421). As such, I remain unconvinced by the discussion that follows. This is problematic given how much time the paper devotes to it.

Unless you can present additional lines of evidence which support your conclusions (e.g. GCM simulations c.f. Liu et al., 2009), these results are highly speculative.

2) TEX86 offsets and high variability

In this paper, the authors show an intriguing temperature offset between two contrasting lithologies. They argue that the laminated carbonate-rich marls reflect glacial cycles, whereas the bioturbated carbonate-rich deposits reflect interglacial cycles. However, I have two concerns:

Firstly, this relies heavily upon Salabarnada et al. which is currently in review.

Secondly, there are other mechanisms which could account for this variability. These include: a) oxic degradation and differential degradation of core GDGTs and/or b) changes in the Thaumarchaeotal community (e.g. deep vs shallow ecotypes).

The latter is particularly important to consider as"...the abundance of 'shallow' versus 'deep water' Thaumarchaeotal communities at deep water sites, like Site U1356, could be affected by the presence of sea ice and the relative influence of (proto-)Component Deep Water upwelling" (lines 220-222).

One way to assess this would be to compare GDGT-2/3 ratios for glacial and interglacial deposits (see Taylor et al., 2013 but also Kim et al., 2015; GCA). You may also want to revisit Littler et al. (2014; P3), as they observe a similar offset in TEX86 values between laminated and homogenous marls during the Cretaceous.

Finally, you also have quite large variations in TEX86 values even within the same lithofacies (e.g ∼31 Myr ago). Is this a true climate signal? Or are there additional controls on the GDGT distribution?

3) Summer Bias

The authors argue that TEX86 values are biased towards summer SST (lines 353-361). Although this observation has also been made for other high-latitude sites during the Oligocene (e.g. ODP Site 511), it is quite speculative and is based upon the assumption that ancient high-latitude GDGT export is similar to the modern. This is quite an assumption! Therefore, do you have any other evidence for a summer bias?

4) Comparison with CO2 records

You observe significant temperature fluctuations in your record (up to 10 °C). This may be partly related to glacial/interglacial variability. However, is there also a potential role

for CO2? (see Zhang et al. (2011) and Anagnostou et al. (2016) for recent Oligocene CO2 estimates). It might be worth showing these CO2 records in Figure 4 too.

5) Branched GDGTs:

Although the authors have analysed branched GDGT (see Supp. Table), MBT/CBT values were not reported. However, this could provide important insights into continental air temperatures during the Oligocene and would be an interesting addition to this paper.

For example, how do MBT'/CBT values compare to TEX86 estimates? Do they exhibit the same temporal trends? Are they offset? Do they max out? etc etc.

6) Calibrations

This study uses the linear calibration of Kim et al. (2010). However, it is important to note that Kim et al. (2010) plots SST on the y-axis (see Kim et al. 2010; Fig.5). As such, this calibration will suffer from a regression dissolution bias and should not be used.

This paper also flips between different calibrations and there needs to be some consistency in the text and figures. For example, in Figure 3 you calculate SSTs at IODP 1356 with TEX86 (Kim 2010) and BAYSPAR (T&T 2015). However, ODP Site 511 is only shown using TEX86 (Kim 2010). Why not also recalculate with BAYSPAR?

Similarly, you only use only BAYSPAR in Figure 4, despite the manuscript stating that Kim et al. (2010) was the preferred calibration (line 241).

Minor comments:

There are a few other TEX86 datasets which might be of interest to the authors;

a. Wade et al, (2012) -> TEX86 values span the earliest Oligocene (ca. 34-32 Ma). b. Zhang et al., (2011) -> TEX86 values span most of the Oligocene.

[Figure]

66: state CO2 estimates for the Oligocene here

93: specifically, ISOPRENOIDAL glycerol dialkyl glycerol tetraethers

94: Careful of the wording here. If you have GDGTs, you will likely have other fossil organics preserved. Do you mean that these are the only fossil "paleothermometers" which are preserved?

101-106: this should probably go into the discussion.

102: BAYSPAR is not strictly a local/ regional calibration. It searches the global core-top dataset for TEX86 values which are similar to the measured value and draws regression parameters from these modern locations.

141: space needed between oceanography and are

140: should this read "Palaeoceanographic setting"?

194: they are not always minor components. For example, in arid and/or alkaline settings they can be the major components (e.g. Huang et al., 2014)

202: why can #ringtetra discriminate between marine and soil-derived GDGTs?

208-220: it has been shown that TEX86-L does not work (e.g. Taylor et al., 2013; Hernandez-Sanchez, 2014) and I think that this discussion can be shortened significantly.

209: these are 'assumed' to originate from Thaumarchaeota. But they can have many sources!

221: Do you have any constraints on water depth? If the site is less than 100m depth, then the presence of "subsurface" archaea is likely to be minimal. However, if the site is >1000m, then subsurface archaea might be an important contribution to the sedimentary GDGT pool.

280: The Methane Index does not flag methanogens. Rather, it can identify anaerobic

methanotrophs (i.e. ANME).

286: What are "non-temperature related influences on TEX86"? It might be useful to go into a little more detail here...

283-284: You state that 8 had high BIT values, but only 5 were discarded? Why?

317: To be clear, BAYSPAR is based upon "581 coretops, including the 426 sites from the Kim et al. (2010) calibration study and an additional 155 sites from regional coretop TEX86 studies (Leider et al., 2010; Shintani et al., 2010; Ho et al., 2011; Wei et al., 2011; Chazen, 2011; Shevenell et al., 2011; Fallet et al., 2012; Jia et al., 2012; Hu et al., 2012)". See Tierney and Tingley (2015) for more details.

319: Can you explain where the 3.5°C error bars come from. I thought that BAYSPAR gave you 90% uncertainty levels rather than a definitive calibration error.

331: What was the paleolatitude of the Wilkes Land section during the Oligocene?

Line 404: how exactly does this resampling work? It is not clear from the text.

Figure 5: Most of the data in Figure 5 is already in Figure 4. As such, I would recommend removing it.

---

## Referee Comment (RC2) · Anonymous Referee #2 · 19 Feb 2018

In this paper Hartman et al., present Southern Ocean TEX86-derived temperature estimates from the Oligocene. They then use the TEX86-derived temperature reconstruction to de-convolve a $\delta$18O of benthic foraminifera from an equatorial Pacific site into the $\delta$18O of seawater component to infer relative stability of the Antarctic ice sheet during the Oligocene. I think the TEX86 record is a useful contribution to our understanding of temperature conditions around Antarctica during the Oligocene. However, I found the "thought experiment" inferring Antarctic ice volume stability (lines 419-420), particularly given the limitations of TEX86, unconvincing. The authors are very mindful about the limitations of this "thought experiment" but inferring ice volume stability feels over-reaching. I think a revised manuscript should focus on the regional oceanography and polar frontal systems. I also found it difficult to comprehensively review this

manuscript because it references two submitted articles (Salabarnada et al., and Bijl et al.,) related to the lithology and surface water conditions.

There are many fundamental assumptions the authors make and explore within the manuscript with respect to relative contribution of temperature and ice volume in the $\delta$18O of benthic foraminifera from Site 1218:

1) Temperature bias in TEX86 is to summer and deep water production (in the modern) is to winter. The authors note this throughout the manuscript but this is a difficult temperature disconnect to constrain. What is the seasonal range in temperatures from summer to winter today? The winter temperatures during deep water formation are constrained because seawater freezes at -2C. Summer temperatures, particularly in a warmer world, could vary by a lot. In particular, the TEX86-derived temperatures are nearly twice that of the Mg/Ca bottom water temperature record.

2) No subsurface temperature bias in TEX86. Given that the temperatures vary by >10C and this assumption relies on a submitted manuscript (Salabarnada et al.,), I found this assumption difficult to evaluate. My main concern is that "interglacial" and "glacial" temperatures are related to lithology. The packaging and flux of TEX86 to the deep ocean is likely very different during these times: "interglacial" temperatures are during bioturbated carbonate-rich periods and "glacial" are during laminated silty periods. Also, are there post-depositional processes that might influence TEX86 estimates due to the change in lithology?

It seems given the uncertainty and variability in the temperature reconstructions, all calibrations should be discussed, including the subsurface ones. The BAYSPAR calibration itself (Tierney and Tingley, 2015), which is discussed in some detail, uses regional factors such as the vertical temperature gradient and related subsurface TEX86 influence to reconstruct temperature.

3) The authors mostly dismiss the Lear et al., 2004 Mg/Ca bottom water temperature record. This is odd because the Mg/Ca record is from the same site as the $\delta$18O of

benthic foraminifera (Site 1218) so it should be discussed in some length. The authors note changes in Mg/Ca of seawater and carbonate ion may influence the Mg/Ca-based temperature reconstruction. There are many uncertainties about the Mg/Ca paleotemperature sensitivity to changes in Mg/Ca of seawater (Evans et al., 2016 and for a nice discussion see Lear et al., 2015 for ice volume estimates for the Miocene to present) but the relative direction in bottom water temperatures shouldn't be an issue. The fact that the TEX86 and Mg/Ca-derived temperature estimates have different trends can't be explained by Mg/Ca of seawater changes. Additionally, the benthic foraminifera used in the Lear et al., 2004 study are an infaunal species, largely insulated from long-term changes in in carbonate ion (Lear et al., 2015, Ford et al., 2016).

4) Given the changes in lithology and the offset between the glacial and interglacial LOESS curves is constant, I'm not sure resampling the "glacial (values above average $\delta$18O) and interglacial (values below average $\delta$18O) $\delta$18O trends at Site 1218" (lines 403-405 is the best approach. In fact, I think much of the discussion in the section "4.3 sea surface temperature variability at glacial and interglacial time scales" is poorly supported given the uncertainty in the age model, lithology, and TEX86-based temperature estimates. A more thoughtful approach would a comparison figure of mean $\delta$18O of seawater estimates from 1) LOESS TEX86 and a LOESS $\delta$18O of benthic foraminifera and 2) the high-resolution Mg/Ca and $\delta$18O of benthic foraminifera.

5) The Site 1218 $\delta$18O of benthic foraminifera is used because it covers the entire record. However, are the trends in $\delta$18O of seawater different when the other high resolution Site 1264 $\delta$18O of benthic foraminifera is used? Any one location can be influenced by changes in hydrography.

Minor comments: The authors should include changes in paleolatitude and whether that might influence the temperature record.

Are there large changes in sedimentation rate that might influence preservation and/or these records?

---

## Referee Comment (RC3) · S. Gallagher (Referee) · 19 Mar 2018

This is very good new organic proxy dataset from offshore Wilkes Land. The authors present a near field palaeotemperature record that although is much lower in resolution compared to other proxy datasets it sheds light for the first time consideration of the long term sea surface temperature evolution of this Wilkes Land margin.

I have made extensive comments and suggestions in the attached annotated text to this paper.

I would like to add the following to the discussion:

Reference to EAIS volume changes in line 70 page 3: As I iterated in my review of the Bjil et al submission to this volume: I appreciate the utility of using isotopes to interpret Antarctic Ice Sheet variability as summarise by Liebrand et al (2017) (www.pnas.org/cgi/doi/10.1073/pnas.1615440114) and this approach is used extensively when discussing the Cenozoic greenhouse icehouse transition. However, there are other sections that have been interpreted using backstripping and stratigraphic data in the Gippsland and New Jersey margins that reflect glacio-eustasy in the Oligocene and relative ice volume (Gallagher et al., 2013), it would be useful to consider the significance of these near field and far field sections in any section reviewing ice volume variability. This paper also considers the apparent instability of the EAIS during the Oligocene and presents a sea level curve with Oi events (Figure 6 in Gallagher et al; at slightly higher resolution that the present study) that bears striking similarity to the temperature curve presented in this paper (Figure 4 in this submission).

Gallagher, S. J., G. Villa, R. N. Drysdale, B. S. Wade, H. Scher, Q. Li, M. W. Wallace, and G. R. Holdgate (2013), A near-field sea level record of East Antarctic Ice Sheet instability from 32 to 27 Myr, Paleoceanography, 28, doi: 10.1029/2012PA002326.

More specific comments are below:

Line 95: The core recovery in the Wilkes Land section is certainly not "complete"

Line 145 page 5: These are modelled plate tectonic reconstructions.

Line 160 reference to Bijl et al paper in Jl Micro to be cited?

Line 315 I agree to a certain extent about the lack of identification of Oi events due the gaps in the record (not unexpected during glacials near Antarctica), however, Oi2 is not near 32 Ma (Figure 4) it is actually near 30 Ma and it is possible there is core of this event in the section (see possible correction of Figure 4).

Pages 13 and 14: This section is very interesting yet requires significant clarification, I have suggested ways to enhance the message and tone down the "speculation" in this section hopefully these suggestions help.

In conclusion, once the text has been clarified and the suggestions considered

this will be another useful addition to the relatively sparsely documented Antarctic (palaeo)climate and oceanographic records.

Please also note the supplement to this comment:
https://www.clim-past-discuss.net/cp-2017-153/cp-2017-153-RC3-supplement.pdf

[Figure]

**Supplement:**

[revised manuscript text omitted]

---

## Author Comment (AC1) · 14 Apr 2018

We would like to thank Referee #1 for his/her extensive and constructive review. The major concern of R1, which is also that of R2, is that our attempt to quantitatively disentangle temperature and ice volume from the benthic $\delta$18O record requires too many assumptions. We understand this concern and agree with the reviewers. We agree that the most important message of this paper is to provide the first long-term Oligocene SST record from close to the Antarctic margin, and we will place the focus on this particular aspect in the revised version of our manuscript. In addition, we will discuss the TEX86-based temperature reconstructions, related uncertainties, and their oceanographic and climate implications in a qualitative way. We will also discuss multiple scenarios that might explain the differences and similarities between the

[Figure]

TEX86-based temperature record, the benthic $\delta$18O record and the Mg/Ca-based bottom temperature record, instead of focusing only on the link between temperature and ice volume. Below, we give a point-to-point response to the comments of R1.

ORIGINAL COMMENT:

1) Temperature and ice volume In Section 4.2, the authors attempt to use TEX86 SST estimates to disentangle temperature and ice volume from the deep-sea delta18O signal. However, there are several caveats to this approach. These include: (1) a possible summer bias in TEX86 estimates, (2) uncertainties in the location of deep-water formation, (3) a "poor age model" (the authors words, not mine!) and (4) low sampling resolution (line 416 to 421). As such, I remain unconvinced by the discussion that follows. This is problematic given how much time the paper devotes to it.

Unless you can present additional lines of evidence which support your conclusions (e.g. GCM simulations c.f. Liu et al., 2009), these results are highly speculative.

REPLY:

We agree that our quantitative estimates of the ice volume effect in d18O records based on independent temperature reconstructions from our TEX86-based SST record involves many assumptions. Although we feel that we had objectively discussed these assumptions and associated speculation in our manuscript, the shared concerns of R1 and R2 on this matter indicate that we should revise this part of the discussion. In our revised manuscript, we will limit the discussion to a qualitative assessment of our new temperature record and its influence on the benthic foraminifer $\delta$18O record. Irrespective of all the assumptions involved, we remain confident that our TEX86 record convincingly shows more profound temperature changes both on orbital and on long-term time scales than previously appreciated (e.g., Lear et al. 2004), in an area that is close to the region of deep-water formation (the Wilkes Land-Adélie Coast margin has been identified as a region of deep-water formation during the Eocene by Huck et al. 2017, Paleoceanography). This would imply that a larger portion of the high-amplitude

benthic $\delta$18O can potentially be explained by southern high-latitude climate change and a smaller fraction by ice volume changes. However, we will refrain from quantifying the amount of variation in the benthic $\delta$18O record that could be explained by our data. Instead, we will focus the discussion on how other aspects, such as the moving of the Southern Ocean polar fronts and the position of deep-water formation, could explain the differences and similarities between the benthic $\delta$18O record, the TEX86-based temperature record, and the Mg/Ca-based bottom-water temperature record.

Numerical model simulations of adequate time interval (i.e. geographical boundary conditions of the Oligocene) and spatial resolution are currently not available, although we have initiated collaboration with modelers to produce such simulations in the near future. We choose to leave the model outcomes for a future paper, and solely focus on the data in this current manuscript.

ORIGINAL COMMENT:

In this paper, the authors show an intriguing temperature offset between two contrasting lithologies. They argue that the laminated carbonate-rich marls reflect glacial cycles, whereas the bioturbated carbonate-rich deposits reflect interglacial cycles. However, I have two concerns: Firstly, this relies heavily upon Salabarnada et al. which is currently in review.

REPLY:

We are aware that it would have been difficult for reviewers to assess the veracity of papers that are still under review. This is also one of the reasons why we chose to submit our papers (Hartman et al., Bijl et al., and Salabarnada et al.) to Climate of the Past, as Copernicus enables all reviewers (in fact everybody) to access papers under review (and join in on those discussions) for the purpose of their own review.

Indeed, the lithological interpretations are not part of this paper and are entirely from Salabarnada et al. Crucially, the interpretation of the lithofacies as being representative

of glacial versus interglacial deposits in Salabarnada et al. are based on the integration of the facies (characterized on the basis of sedimentological data, physical properties, and geochemical data) independently of our TEX86 results, and merely stem from bottom current and pelagic sedimentation variations. We want to point out that Salabarnada et al. already submitted replies to their reviews (see the reviews and rebuttals at https://www.clim-past-discuss.net/cp-2017-152/#discussion). The consistent offset between TEX86-based SSTs between the lithologies supports the independent interpretation of the lithofacies by Salabarnada et al. very nicely. This is to us additional support for a SST signal preserved in TEX86.

ORIGINAL COMMENT:

Secondly, there are other mechanisms which could account for this variability. These include: a) oxic degradation and differential degradation of core GDGTs and/or b) changes in the Thaumarchaeotal community (e.g. deep vs shallow ecotypes). The latter is particularly important to consider as": : :the abundance of 'shallow' versus 'deep water' Thaumarchaeotal communities at deep water sites, like Site U1356, could be affected by the presence of sea ice and the relative influence of (proto-)Component Deep Water upwelling" (lines 220-222). One way to assess this would be to compare GDGT-2/3 ratios for glacial and interglacial deposits (see Taylor et al., 2013 but also Kim et al., 2015; GCA). You may also want to revisit Littler et al. (2014; P3), as they observe a similar offset in TEX86 values between laminated and homogenous marls during the Cretaceous.

REPLY:

a) The effect of oxic degradation has been studied by Huguet et al. (2009, OG) on turbidites from the Madeira Abyssal Plain. Although differences between TEX86 values from unoxidized and oxidized sediments have been observed, these are not consistent and it seems therefore that differential degradation of isoprenoid GDGTs does not play a role (Huguet et al. 2009, OG). Instead, oxic degradation may lead to an increased

relative influence of soil-derived isoprenoid GDGTs, which could bias the TEX86 in different ways depending on the composition of the soil-derived isoprenoid GDGTs (Huguet et al. 2009, OG). An increased relative contribution of soil-derived organic matter to marine sediments can be identified using the BIT Index (Hopmans et al., 2004; Weijers et al., 2006). Following this approach, we have discarded nine samples with a BIT Index above 0.3, indicating that the terrestrial contribution could potentially have affected TEX86 values. We acknowledge that this is currently not mentioned in the manuscript and we will add this discussion in a revised version of the manuscript.

b) GDGT2/3 values are available in Table S1 in the Supplementary Information. For the revised manuscript, we will add a figure displaying the GDGT2/3 ratios to facilitate easy comparison between glacial and interglacial lithologies. Although there is considerable variability in GDGT2/3 ratios throughout the record (which is why we refrain from using TEX86-L), it cannot explain the differences in TEX86 between glacial and interglacial sediments. Nor can it explain the long-term trends in our data.

ORIGINAL COMMENT:

Finally, you also have quite large variations in TEX86 values even within the same lithofacies (e.g _31 Myr ago). Is this a true climate signal? Or are there additional controls on the GDGT distribution?

REPLY:

Considering that we have excluded all known biases due to soil-derived input, methanogenic and methanotrophic input, and oxic degradation, and the fact that there is no relation between the TEX86 values and the GDGT2/3 ratio, we interpret the variation in TEX86 as a true temperature (climate) signal. We agree that there is indeed quite some variation within the bioturbated facies around 31 Ma, and will provide and discuss possible explanations for this variability in the revised version of the manuscript.

ORIGINAL COMMENT:

[Figure]

3) Summer Bias The authors argue that TEX86 values are biased towards summer SST (lines 353-361). Although this observation has also been made for other high-latitude sites during the Oligocene (e.g. ODP Site 511), it is quite speculative and is based upon the assumption that ancient high-latitude GDGT export is similar to the modern. This is quite an assumption! Therefore, do you have any other evidence for a summer bias?

REPLY:

The potential summer bias in high latitude TEX86-based SST reconstructions has been discussed extensively in other papers (e.g., Sluijs et al., 2008; Bijl et al., 2009; 2010; 2013). Like in these warm past climates, we expect that primary productivity in the Oligocene Southern Ocean was in sync with seasonal availability of light irrespective of the presence of sea ice. Transport of isoGDGTs likely requires fecal pelleting to sink effectively to the sea floor, and therefore depends on the presence of larger zooplankton that feed on the phytoplankton. This means that, like today, TEX86 temperature reconstructions are likely skewed towards the season with highest primary productivity, i.e. the summer. We will explain this more clearly in a new version of the manuscript.

ORIGINAL COMMENT:

4) Comparison with CO2 records You observe significant temperature fluctuations in your record (up to 10 _C). This may be partly related to glacial/interglacial variability. However, is there also a potential role for CO2? (see Zhang et al. (2011) and Anagnostou et al. (2016) for recent Oligocene CO2 estimates). It might be worth showing these CO2 records in Figure 4 too.

REPLY:

We agree with the reviewer that atmospheric CO2 concentrations were likely the driving factor of Oligocene glacial-interglacial variability (DeConto et al. 2008; Liebrand et al. 2017). However, locally, surface water temperature variability can respond very

sensitively to glacial-interglacial climate change and in a non-linear way, e.g., via the migration of ocean fronts. The discussion about the forcing mechanism for Oligocene high-latitude glacial-interglacial climate variability lies beyond the scope of this paper. Moreover, the resolution of the existing $CO_2$ reconstructions for the Oligocene (Zhang et al. 2013) (as well as those for the Eocene (Anagnostou et al. 2016)) does not capture the glacial-interglacial variability seen in our record. We will therefore refrain from adding a $CO_2$ reconstruction to Figure 4, as it would suggest a correlation between the two, which we cannot say based on our results.

ORIGINAL COMMENT:

5) Branched GDGTs: Although the authors have analysed branched GDGT (see Supp. Table), MBT/CBT values were not reported. However, this could provide important insights into continental air temperatures during the Oligocene and would be an interesting addition to this paper.

For example, how do MBT'/CBT values compare to TEX86 estimates? Do they exhibit the same temporal trends? Are they offset? Do they max out? etc etc.

REPLY:

Although we agree that data on Oligocene air temperatures would be a great addition to our understanding of Oligocene Antarctic climate evolution, we fear that branched GDGT (brGDGT) data obtained from Site U1356 are unable to provide reliable information on this front. The reason for this is the very low absolute and relative abundances of soil-derived brGDGTs (BIT<0.3), which is probably caused by the too large distance to shore and/or limited soil development and subsequent transport from the land. In addition, a substantial portion of the samples analyzed has #rings-tetra values higher than 0.4 (see Table S1), which indicates that temperature estimates for these samples are likely affected by a contribution of in situ produced brGDGTs (Sinninghe Damsté, 2016). All of the above makes it difficult to infer a reliable long-term atmospheric temperature trend at this stage.

ORIGINAL COMMENT:

6) Calibrations This study uses the linear calibration of Kim et al. (2010). However, it is important to note that Kim et al. (2010) plots SST on the y-axis (see Kim et al. 2010; Fig.5). As such, this calibration will suffer from a regression dissolution bias and should not be used.

REPLY:

Referee #1 is correct in stating that the calibration of Kim et al. (2010) suffers from a regression dilution bias caused by the uncertainty in the measured TEX86 values plotted on the x-axis, and we shall add this potential complication in the methods section. This bias causes flattening of the slope (Hutcheon et al. 2010, BMJ) and therefore affects TEX86-based temperature reconstructions at the lower and upper end of the calibration range. However, because TEX86 in our record lies around the middle of the total TEX86 range used for the linear calibration of Kim et al. (2010), reconstructed temperature values for Site U1356 will not be severely biased.

Indeed, the BAYSPAR method is the only TEX86 calibration that is not affected by the regression dilution. However, by applying the BAYSPAR method, only core tops within a selection of 20°x20° grid boxes are used, thereby excluding all of the low-temperature Southern Ocean core-top calibration values (Fig. 3). Considering that Site U1356A is a high-latitude Southern Ocean site, we did not want to only use BAYSPAR and, therefore, plotted the linear calibration of Kim et al. (2010) for comparison. Although this linear calibration shows large scatter at the lower temperature end, it has been shown that all Southern Ocean core-top values in the Kim et al. (2010) dataset fall within the standard error ($\pm 5.2$°C) of the linear calibration of Kim et al. (2010) (Ho et al. 2014, GCA). In addition, the fact that the temperature reconstruction based on the Kim et al. (2010) calibration compares very well with the BAYSPAR-based temperature record (Fig. 3) indicates that the effect of regression dilution bias is, in our case, relatively minimal.

In the revised manuscript we will clarify our choice to use both the BAYSPAR and the Kim et al. (2010) linear calibration.

ORIGINAL COMMENT:

This paper also flips between different calibrations and there needs to be some consistency in the text and figures. For example, in Figure 3 you calculate SSTs at IODP 1356 with TEX86 (Kim 2010) and BAYSPAR (T&T 2015). However, ODP Site 511 is only shown using TEX86 (Kim 2010). Why not also recalculate with BAYSPAR?

REPLY:

We agree that there are some inconsistencies in the use of calibrations in the text and figures. We shall recalculate the temperatures of ODP Site 511 using BAYSPAR and add them to Figure 3.

ORIGINAL COMMENT:

Similarly, you only use only BAYSPAR in Figure 4, despite the manuscript stating that Kim et al. (2010) was the preferred calibration (line 241).

REPLY:

To clarify, we do not state that we prefer the linear calibration of Kim et al. (2010) over the BAYSPAR calibration. This is why we also apply the BAYSPAR calibration. The offset between these two calibrations is, however, only 0.5°C, well within the calibration error of both regressions. Because of the small temperature difference between the two calibrations it is only for convenience that we showed only one calibration in Figure 4. In the revised manuscript we will plot both reconstructions in Figure 4.

ORIGINAL COMMENT:

Minor comments:

There are a few other TEX86 datasets which might be of interest to the authors;

a. Wade et al, (2012) -> TEX86 values span the earliest Oligocene (ca. 34-32 Ma). b. Zhang et al., (2011) -> TEX86 values span most of the Oligocene.

REPLY:

TEX86 data of Wade et al. (2012), and Zhang et al. (2013, not 2011) clearly show that the early Oligocene did experience a latitudinal temperature gradient. Although a valid observation in itself, these low-latitude regions are not the focus of our study. Our study focusses on near-Antarctic SST estimates and trends and how these relate to our current knowledge of the Oligocene Antarctic ice-sheet dynamics.

ORIGINAL COMMENT:

66: state CO2 estimates for the Oligocene here

REPLY:

We will state CO2 estimates for the Oligocene

ORIGINAL COMMENT:

93: specifically, ISOPRENOIDAL glycerol dialkyl glycerol tetraethers

REPLY:

We will insert 'isoprenoidal'

ORIGINAL COMMENT:

94: Careful of the wording here. If you have GDGTs, you will likely have other fossil organics preserved. Do you mean that these are the only fossil "paleothermometers" which are preserved?

REPLY:

We mean specifically fossil paleothermometers.

ORIGINAL COMMENT:

101-106: this should probably go into the discussion.

REPLY:

We agree with the reviewer that this should not be part of the introduction. This section will be transferred to either the methods section 2.6 on TEX86 calibrations or the discussion of the revised manuscript.

ORIGINAL COMMENT:

102: BAYSPAR is not strictly a local/ regional calibration. It searches the global coretop dataset for TEX86 values which are similar to the measured value and draws regression parameters from these modern locations.

REPLY:

We agree that BAYSPAR is not strictly a local/regional calibration. By selecting only TEX86 core-top values similar to the ones measured in Oligocene samples at Site U1356 BAYSPAR constructs a calibration based on multiple modern-day analogue sites and therefore is not regional. We shall therefore be more careful using the word 'regional'.

ORIGINAL COMMENT:

141: space needed between oceanography and are

REPLY:

A space will be added

ORIGINAL COMMENT:

140: should this read "Palaeoceanographic setting"?

REPLY:

Paleoceanographic setting would be a better heading.

ORIGINAL COMMENT:

194: they are not always minor components. For example, in arid and/or alkaline settings they can be the major components (e.g. Huang et al., 2014)

REPLY:

Indeed, isoGDGTs are major components in arid/alkaline soils when compared to brGDGTs. However, compared to in marine sediments, the concentration of isoGDGTs in soils is always low. We shall therefore not alter this statement.

ORIGINAL COMMENT:

202: why can #ringtetra discriminate between marine and soil-derived GDGTs?

REPLY:

This information is not explicitly given in the manuscript and we shall include this in the revised version. To answer the question, #rings-tetra can discriminate between marine and soil-derived branched GDGTs as the composition of soil-derived GDGTs typically show high amounts of the acyclic tetramethylated GDGT-Ia, while a dominance of cyclic tetramethylated (Ib, Ic) (and also pentamethylated) brGDGTs has been attributed to in situ production within the sediments (Sinninghe Damsté 2016).

ORIGINAL COMMENT:

208-220: it has been shown that TEX86-L does not work (e.g. Taylor et al., 2013; Hernandez-Sanchez, 2014) and I think that this discussion can be shortened significantly.

REPLY:

Agreed. A less detailed discussion is sufficient by stating earlier that TEX86L does not work due to its sensitivity to changes in the GDGT2/3 ratio as shown by Hernandez-Sanchez et al. (2014).

ORIGINAL COMMENT:

209: these are 'assumed' to originate from Thaumarchaeota. But they can have many sources!

REPLY:

We shall change the sentence to "In marine sediments these lipids are assumed to originate mostly from cell membranes of marine Thaumarchaeota, because they are one of the dominant prokaryotes in today's ocean and occur throughout the entire water column".

ORIGINAL COMMENT:

221: Do you have any constraints on water depth? If the site is less than 100m depth, then the presence of "subsurface" archaea is likely to be minimal. However, if the site is >1000m, then subsurface archaea might be an important contribution to the sedimentary GDGT pool.

REPLY:

Today the water depth of Site U1356 is 3992 m (see material & methods section). We have no quantitative constraints on water depth during the Oligocene, but the sediments, characterized by hemipelagic deposits reworked by bottom currents and distal gravity flows, as well as biota suggest a deep-water setting in the Oligocene (Escutia et al., 2014 Develop. Mar. Geol.). It is therefore certainly not a shallow (<1000 m depth) site. Regardless, it is more likely that GDGTs derived from shallower waters (<1000 m depth) are more effectively transported to the sediments through a.o. fecal pelleting (Schouten et al., 2013 and references cited therein). If deep archaeal communities would have contributed to the sedimentary GDGT pool, this would result in a higher GDGT2/3 ratio for these samples, which will have only a minor effect on the TEX86 (i.e. TEX86H) index (Hernández-Sánchez et al. 2014).

ORIGINAL COMMENT:

Interactive
comment

280: The Methane Index does not flag methanogens. Rather, it can identify anaerobic methanotrophs (i.e. ANME).

REPLY:

In line 280 proxies for identifying methanotrophs have mistakenly been lumped together with those identifying methanogens. We shall correct for this.

ORIGINAL COMMENT:

286: What are "non-temperature related influences on TEX86"? It might be useful to go into a little more detail here...

REPLY:

We will elaborate more on 'non-temperature related influences on TEX86', such as archaeal growth rates or an input from methanogenic Euryarchaeota (Zhang et al., 2016 Paleoceanography), in the revised manuscript.

ORIGINAL COMMENT:

283-284: You state that 8 had high BIT values, but only 5 were discarded? Why?

REPLY:

We meant to say that in addition to the eight samples with high BIT values, another 5 were discarded based on high GDGT-0/Cren or Methane Index values. As all samples with too high BIT values also had high GDGT-0/Cren or Methane Index values, a total of 13 samples showed indications of input from methanogens or methanotrophs. We will clarify this in the revised version.

ORIGINAL COMMENT:

317: To be clear, BAYSPAR is based upon "581 coretops, including the 426 sites from the Kim et al. (2010) calibration study and an additional 155 sites from regional coretop TEX86 studies (Leider et al., 2010; Shintani et al., 2010; Ho et al., 2011; Wei et al.,

2011; Chazen, 2011; Shevenell et al., 2011; Fallet et al., 2012; Jia et al., 2012; Hu et al., 2012)". See Tierney and Tingley (2015) for more details.

REPLY:

We shall mention the additional 155 sites aside from the Kim et al. (2010) calibration set.

ORIGINAL COMMENT:

319: Can you explain where the 3.5°C error bars come from. I thought that BAYSPAR gave you 90% uncertainty levels rather than a definitive calibration error.

REPLY:

Referee #1 is correct in stating that the BAYSPAR program gives you 90% confidence levels for each sample. Assuming that the error is normally distributed around the mean, the lower and upper 90% confidence interval boundary can be calculated as the mean plus or minus 1.645 times the standard error. As for each sample the mean, upper and lower confidence interval are known, the standard error can be calculated. On average the calculated standard error for all of the samples is $\pm4.2°C$. The $\pm3.5°C$ standard error was miscalculated based on BAYSPAR giving the 95% confidence levels. We shall make this calculation in the methods section and correct this error.

ORIGINAL COMMENT:

331: What was the paleolatitude of the Wilkes Land section during the Oligocene?

REPLY:

We shall add the paleolatitude of Site U1356 to the manuscript, which is 58.86°S.

ORIGINAL COMMENT:

Line 404: how exactly does this resampling work? It is not clear from the text.

REPLY:

Resampling is done by predicting the $\delta$18O value from a LOESS fit through the glacial and interglacial $\delta$18O values. We shall explain this in the text.

ORIGINAL COMMENT:

Figure 5: Most of the data in Figure 5 is already in Figure 4. As such, I would recommend removing it.

REPLY:

We shall consider removing this figure. However, the variance around the mean is not always clearly visible when the long-term trend is not removed.

---

## Author Comment (AC3) · 14 Apr 2018

We thank Stephen Gallagher for reviewing our manuscript and for acknowledging the value of our dataset. His annotations to our manuscript showed us that some sections lack clarity. In particular the last part of section 4.2, which involves the "thought experiment", and section 4.3 that discusses the reconstructed temperature variability. Also, considering the comments of Referees #1 and #2, we have decided to significantly restructure those sections, placing more emphasis on the role of paleoceanography (polar fronts), and to refrain from quantifying ice volume changes. In the revised manuscript we will discuss several scenarios that can explain the differences and similarities between our TEX86-based temperature record, the benthic $\delta$18O records and the Mg/Ca-based bottom-water temperature record. We believe that this approach will

improve the structure and clarity of the manuscript.

ORIGINAL COMMENT:

This is very good new organic proxy dataset from offshore Wilkes Land. The authors present a near field palaeotemperature record that although is much lower in resolution compared to other proxy datasets it sheds light for the first time consideration of the long term sea surface temperature evolution of this Wilkes Land margin.

I have made extensive comments and suggestions in the attached annotated text to this paper.

REPLY:

All comments and suggestions in the annotated text were clear, mostly they were related to the choice of words or to incorrect English and suggestions will be included in a revised manuscript. The reviewer's major comments are addressed below.

Comment: In the annotated manuscript, in line 238, the reviewer asks for clarification on why TEX86 would not have been influenced by subsurface temperatures in the absence of low-salinity waters due to sea ice melt.

Answer: It has been shown that the sea-ice influenced, low-salinity surface waters of today's Southern Ocean contain virtually no Thaumarchaeota in the top layer of the water column (0-45 meter below sea level (mbsl), Kalanetra et al. 2009 Environ. Microbiol.). Instead, the GDGTs are derived from Thaumarchaeota in the deeper water column (45-105 mbsl, Kalanetra et al., 2009 Environ. Microbiol.) and, therefore, TEX86 does not represent a true surface water signal (see also Kim et al., 2012 Geophys. Res. Let.). Dinoflagellate cysts in the same sediments suggest that oceanographic conditions were similar to today's Subtropical Front and that no sea ice was present. Hence, we conclude that TEX86 at Site U1356 does reflect a surface water temperature. We will clarify this in a new version.

Comment: In line 251 Stephen Gallagher has placed a question mark at the word 'prior'

in the sentence "The prior for site U1356 is obtained from recent clumped isotope measurements ($\Delta$47) on planktonic foraminifers from Maud Rise (ODP Site 689) (Petersen and Schrag, 2015), which show early Oligocene temperatures of 12°C."

Answer: the BAYSPAR method is based on Bayesian inference and therefore requires a prior distribution of temperature (i.e. the prior) in order to predict sea surface temperatures from the observed TEX86 values. In general, the prior is our initial belief or scientific understanding of the unknown quantities to be estimated, in this case sea surface temperature (Tierney & Tingley, 2014). As for deep time temperature reconstruction this prior cannot be based on modern-day annual mean sea surface temperatures, the BAYSPAR method requires a user-specified mean and variance for this prior (Tierney & Tingley, 2014). Therefore, we use previous estimates of southern high-latitude early Oligocene seawater temperatures (12°C based on $\Delta$47) as a prior for our TEX86-based sea surface temperature reconstruction. We will better explain this in the revised manuscript.

ORIGINAL COMMENT:

I would like to add the following to the discussion: Reference to EAIS volume changes in line 70 page 3: As I iterated in my review of the Bjil et al submission to this volume: I appreciate the utility of using isotopes to interpret Antarctic Ice Sheet variability as summarise by Liebrand et al (2017) (www.pnas.org/cgi/doi/10.1073/pnas.1615440114) and this approach is used extensively when discussing the Cenozoic greenhouse icehouse transition. However, there are other sections that have been interpreted using backstripping and stratigraphic data in the Gippsland and New Jersey margins that reflect glacio-eustasy in the Oligocene and relative ice volume (Gallagher et al., 2013), it would be useful to consider the significance of these near field and far field sections in any section reviewing ice volume variability. This paper also considers the apparent instability of the EAIS during the Oligocene and presents a sea level curve with Oi events (Figure 6 in Gallagher et al; at slightly higher resolution that the present study) that bears striking similarity to the temperature curve presented in this paper (Figure 4

in this submission).

Gallagher, S. J., G. Villa, R. N. Drysdale, B. S. Wade, H. Scher, Q. Li, M. W. Wallace, and G. R. Holdgate (2013), A near-field sea level record of East Antarctic Ice Sheet instability from 32 to 27 Myr, Paleoceanography, 28, doi: 10.1029/2012PA002326.

REPLY:

We agree that the sea level curve reconstructed in this paper shows the same long-term trends as our temperature record. In our new version of the manuscript, we will sketch several scenarios to explain the differences and similarities between our TEX86 record and the benthic $\delta$18O record in a more qualitative way. As this will likely involve ice volume changes and therefore global sea level changes, the paper by Gallagher et al. (2013) will be a nice addition to this discussion, providing a framework for our theories.

ORIGINAL COMMENT:

More specific comments are below:

Line 95: The core recovery in the Wilkes Land section is certainly not "complete"

REPLY:

The reviewer is correct in this and we will revise the sentences where we give the impression that Hole U1356A is without hiatuses.

ORIGINAL COMMENT:

Line 145 page 5: These are modelled plate tectonic reconstructions.

REPLY:

We will correct this. Indeed tectonic reconstructions show that Australia and South America were closer to Antarctica. We meant to say that numerical modeling of ocean currents shows that the strength of the circum-polar current was limited by these narrow

gateways (Hill et al. 2013).

ORIGINAL COMMENT:

Line 160 reference to Bijl et al paper in Jl Micro to be cited?

REPLY:

We will cite Bijl et al. (2018, J. Micropal.) at the end of this line to refer to the position of M. escutiana.

ORIGINAL COMMENT:

Line 315 I agree to a certain extent about the lack of identification of Oi events due the gaps in the record (not unexpected during glacials near Antarctica), however, Oi2 is not near 32 Ma (Figure 4) it is actually near 30 Ma and it is possible there is core of this event in the section (see possible correction of Figure 4).

REPLY:

We will correct for this mistake and place Oi-2 near 30 Ma in Figure 4. Although it might be possible that Oi-2 is recorded in U1356A, age model limitations prevent us to be certain.

ORIGINAL COMMENT

Pages 13 and 14: This section is very interesting yet requires significant clarification, I have suggested ways to enhance the message and tone down the "speculation" in this section hopefully these suggestions help.

REPLY:

We will reduce the speculation by refraining from a quantitative comparison between TEX86 and benthic $\delta$18O.

ORIGINAL COMMENT:

In conclusion, once the text has been clarified and the suggestions considered this will be another useful addition to the relatively sparsely documented Antarctic (palaeo)climate and oceanographic records.

REPLY:

We thank the reviewer for these positive comments.

———————————————————

---

## Author Response (AR1)

**Universiteit Utrecht**

**Editorial Board, Climate of the Past**

To the editorial Board

**Faculty of Geosciences,
Department of Earth Sciences**
Marine Palynology and
Paleoceanography

**Visitors Address**
Princetonlaan 8a
3584 CB Utrecht
The Netherlands

**Your reference**
**Our reference**   Climate of the Past manuscript
**Phone**   +31 30 253 2630
**Email**   J.D.Hartman@uu.nl
**Website**   https://www.uu.nl/staff/JDHartman
**Date**   November 22, 2017
**Subject**   Submission of manuscript to Climate of the Past (cp-2017-153)

Dear David Thornalley,

We would like to thank you for the extra week time granted for resubmitting this manuscript. Please find enclosed a final and marked-up version of our manuscript, entitled "**Oligocene and Miocene TEX$_{86}$-derived seawater temperatures from offshore Wilkes Land (East Antarctica)** ", as well as the rebuttals to the reviewers.

Firstly, as you may have noticed, we have changed the title to include Miocene for the reason we explain below.

We have significantly revised our manuscript, taking into account the comments of the reviewers. Notably, the so-called thought experiment that deduced Antarctic ice volume changes based on our sea surface temperature record was (properly) considered too far-fetched by all reviewers, and has therefore been removed. Instead, the discussion focuses on comparing our sea surface temperature record with existing sea surface temperature, bottom-water temperature and stable oxygen isotope records in a much more in a qualitative way. Trends seen in our temperature records are now interpreted in terms of paleoceanographic changes rather than focused exclusively on ice sheet stability.

As this manuscript is part of triple submission, with the other manuscripts also including data on the Miocene section of the study site (IODP Site U1356), we believed it was necessary to update this manuscript by including recently published Miocene TEX$_{86}$-based sea surface temperature data of Site U1356 (Sangiorgi et al., 2018, Nature Communications). As for the Oligocene data, the relation between TEX$_{86}$ and the lithology was examined, and the reconstructed sea surface temperatures are discussed in relation to existing sea surface temperature, bottom-water temperature and stable oxygen isotope records.

As a final remark, Figure 5 was removed. Henk Brinkhuis was added as a co-author to this manuscript, due to his participation as co-chief to IODP Expedition 318 where the material comes from and his valuable contribution to this new version of the manuscript.

The manuscript contains 4 figures, 2 supplementary figures and 2 supplementary tables and about 10,000 words.

In anticipation of your reply,

Best regards, also on behalf of my co-authors,

Julian D. Hartman
Corresponding author

**Point-by-point response to the reviewers**

**Reply to Referee #1**

We would like to thank Referee #1 for his/her extensive and constructive review.

The major concern of R1, which is also that of R2, is that our attempt to quantitatively disentangle temperature and ice volume from the benthic δ18O record requires too many assumptions. We understand this concern and agree with the reviewers. We agree that the most important message of this paper is to provide the first long-term Oligocene SST record from close to the Antarctic margin, and we will place the focus on this particular aspect in the revised version of our manuscript.

In addition, we will discuss the TEX86-based temperature reconstructions, related uncertainties, and their oceanographic and climate implications in a qualitative way. We will also discuss multiple scenarios that might explain the differences and similarities between the TEX86-based temperature record, the benthic δ18O record and the Mg/Ca-based bottom temperature record, instead of focusing only on the link between temperature and ice volume. Below, we give a point-to-point response to the comments of R1.

ORIGINAL COMMENT:
1) Temperature and ice volume

In Section 4.2, the authors attempt to use TEX86 SST estimates to disentangle temperature and ice volume from the deep-sea delta18O signal. However, there are several caveats to this approach. These include: (1) a possible summer bias in TEX86 estimates, (2) uncertainties in the location of deep-water formation, (3) a "poor age model" (the authors words, not mine!) and (4) low sampling resolution (line 416 to 421). As such, I remain unconvinced by the discussion that follows. This is problematic given how much time the paper devotes to it.

Unless you can present additional lines of evidence which support your conclusions (e.g. GCM simulations c.f. Liu et al., 2009), these results are highly speculative.

REPLY:
We agree that our quantitative estimates of the ice volume effect in d18O records based on independent temperature reconstructions from our TEX86-based SST record involves many assumptions. Although we feel that we had objectively discussed these assumptions and associated speculation in our manuscript, the shared concerns of R1 and R2 on this matter indicate that we should revise this part of the discussion. In our revised manuscript, we will limit the discussion to a qualitative assessment of our new temperature record and its influence on the benthic foraminifer δ18O record. Irrespective of all the assumptions involved, we remain confident that our TEX86 record convincingly shows more profound temperature changes both on orbital and on long-term time scales than previously appreciated (e.g., Lear et al. 2004), in an area that is close to the region of deep-water formation (the Wilkes Land-Adélie Coast margin has been identified as a region of deep-water formation during the Eocene by Huck et al. 2017, Paleoceanography). This would imply that a larger portion of the

high-amplitude benthic δ18O can potentially be explained by southern high-latitude climate change and a smaller fraction by ice volume changes. However, we will refrain from quantifying the amount of variation in the benthic δ18O record that could be explained by our data. Instead, we will focus the discussion on how other aspects, such as the moving of the Southern Ocean polar fronts and the position of deep-water formation, could explain the differences and similarities between the benthic δ18O record, the TEX86-based temperature record, and the Mg/Ca-based bottom-water temperature record.

Numerical model simulations of adequate time interval (i.e. geographical boundary conditions of the Oligocene) and spatial resolution are currently not available, although we have initiated collaboration with modelers to produce such simulations in the near future. We choose to leave the model outcomes for a future paper, and solely focus on the data in this current manuscript.

ORIGINAL COMMENT:
In this paper, the authors show an intriguing temperature offset between two contrasting lithologies. They argue that the laminated carbonate-rich marls reflect glacial cycles, whereas the bioturbated carbonate-rich deposits reflect interglacial cycles. However, I have two concerns:

Firstly, this relies heavily upon Salabarnada et al. which is currently in review.

REPLY:
We are aware that it would have been difficult for reviewers to assess the veracity of papers that are still under review. This is also one of the reasons why we chose to submit our papers (Hartman et al., Bijl et al., and Salabarnada et al.) to Climate of the Past, as Copernicus enables all reviewers (in fact everybody) to access papers under review (and join in on those discussions) for the purpose of their own review.

Indeed, the lithological interpretations are not part of this paper and are entirely from Salabarnada et al. Crucially, the interpretation of the lithofacies as being representative of glacial versus interglacial deposits in Salabarnada et al. are based on the integration of the facies (characterized on the basis of sedimentological data, physical properties, and geochemical data) independently of our TEX86 results, and merely stem from bottom current and pelagic sedimentation variations. We want to point out that Salabarnada et al. already submitted replies to their reviews (see the reviews and rebuttals at https://www.clim-past-discuss.net/cp-2017-152/#discussion). The consistent offset between TEX86-based SSTs between the lithologies supports the independent interpretation of the lithofacies by Salabarnada et al. very nicely. This is to us additional support for a SST signal preserved in TEX86.

ORIGINAL COMMENT:
Secondly, there are other mechanisms which could account for this variability. These include: a) oxic degradation and differential degradation of core GDGTs and/or b) changes in the Thaumarchaeotal community (e.g. deep vs shallow ecotypes).

The latter is particularly important to consider as": : :the abundance of 'shallow' versus 'deep water' Thaumarchaeotal communities at deep water sites, like Site U1356, could be affected by the presence of sea ice and the relative influence of (proto-)Component Deep Water upwelling" (lines 220-222).

One way to assess this would be to compare GDGT-2/3 ratios for glacial and interglacial deposits (see Taylor et al., 2013 but also Kim et al., 2015; GCA). You may also want to revisit Littler et al. (2014; P3), as they observe a similar offset in TEX86 values between laminated and homogenous marls during the Cretaceous.

REPLY:
a) The effect of oxic degradation has been studied by Huguet et al. (2009, OG) on turbidites from the Madeira Abyssal Plain. Although differences between TEX86 values from unoxidized and oxidized sediments have been observed, these are not consistent and it seems therefore that differential degradation of isoprenoid GDGTs does not play a role (Huguet et al. 2009, OG). Instead, oxic degradation may lead to an increased relative influence of soil-derived isoprenoid GDGTs, which could bias the TEX86 in different ways depending on the composition of the soil-derived isoprenoid GDGTs (Huguet et al. 2009, OG). An increased relative contribution of soil-derived organic matter to marine sediments can be identified using the BIT Index (Hopmans et al., 2004; Weijers et al., 2006). Following this approach, we have discarded nine samples with a BIT Index above 0.3, a threshold at which the terrestrial contribution could potentially have affected TEX86 values. We acknowledge that this is currently not mentioned in the manuscript and we will add this discussion in a revised version of the manuscript.

b) GDGT2/3 values are available in Table S1 in the Supplementary Information. For the revised manuscript, we will add a figure displaying the GDGT2/3 ratios to facilitate easy comparison between glacial and interglacial lithologies. Although there is considerable variability in GDGT2/3 ratios throughout the record (which is why we refrain from using TEX86-L), it cannot explain the differences in TEX86 between glacial and interglacial sediments. Nor can it explain the long-term trends in our data.

ORIGINAL COMMENT:
Finally, you also have quite large variations in TEX86 values even within the same lithofacies (e.g _31 Myr ago). Is this a true climate signal? Or are there additional controls on the GDGT distribution?

REPLY:
Considering that we have excluded all known biases due to soil-derived input, methanogenic and methanotrophic input, and oxic degradation, and the fact that there is no relation between the TEX86 values and the GDGT2/3 ratio, we interpret the variation in TEX86 as a true temperature (climate) signal. We agree that there is indeed quite some variation within the bioturbated facies around 31 Ma, and will provide and discuss possible explanations for this variability in the revised version of the manuscript.

ORIGINAL COMMENT:
3) Summer Bias

The authors argue that TEX86 values are biased towards summer SST (lines 353-361). Although this observation has also been made for other high-latitude sites during the Oligocene (e.g. ODP Site 511), it is quite speculative and is based upon the assumption that ancient high-latitude GDGT export is similar to the modern. This is quite an assumption! Therefore, do you have any other evidence for a summer bias?

REPLY:

The potential summer bias in high latitude TEX86-based SST reconstructions has been discussed extensively in other papers (e.g., Sluijs et al., 2008; Bijl et al., 2009; 2010; 2013). Like in these warm past climates, we expect that primary productivity in the Oligocene Southern Ocean was in sync with seasonal availability of light irrespective of the presence of sea ice. Transport of isoGDGTs likely requires fecal pelleting to sink effectively to the sea floor, and therefore depends on the presence of larger zooplankton that feed on the phytoplankton. This means that, like today, TEX86 temperature reconstructions are likely skewed towards the season with highest primary productivity, i.e. the summer. We will explain this more clearly in a new version of the manuscript.

ORIGINAL COMMENT:
4) Comparison with CO2 records

You observe significant temperature fluctuations in your record (up to 10 _C). This may be partly related to glacial/interglacial variability. However, is there also a potential role for CO2? (see Zhang et al. (2011) and Anagnostou et al. (2016) for recent Oligocene CO2 estimates). It might be worth showing these CO2 records in Figure 4 too.

REPLY:

We agree with the reviewer that atmospheric CO2 concentrations were likely the driving factor of Oligocene glacial-interglacial variability (DeConto et al. 2008; Liebrand et al. 2017). However, locally, surface water temperature variability  can respond very sensitively to glacial-interglacial climate change and in a non-linear way, e.g., via the migration of ocean fronts. The discussion about the forcing mechanism for Oligocene high-latitude glacial-interglacial climate variability lies beyond the scope of this paper. Moreover, the resolution of the existing CO2 reconstructions for the Oligocene (Zhang et al. 2013) (as well as those for the Eocene (Anagnostou et al. 2016)) does not capture the glacial-interglacial variability seen in our record. We will therefore refrain from adding a CO2 reconstruction to Figure 4, as it would suggest a correlation between the two, which we cannot say based on our results.

ORIGINAL COMMENT:
5) Branched GDGTs:

Although the authors have analysed branched GDGT (see Supp. Table), MBT/CBT values were not reported. However, this could provide important insights into continental air temperatures during the Oligocene and would be an interesting addition to this paper.

For example, how do MBT'/CBT values compare to TEX86 estimates? Do they exhibit the same temporal trends? Are they offset? Do they max out? etc etc.

REPLY:
Although we agree that data on Oligocene air temperatures would be a great addition to our understanding of Oligocene Antarctic climate evolution, we fear that branched GDGT (brGDGT) data obtained from Site U1356 are unable to provide reliable information on this front. The reason for this is

the very low absolute and relative abundances of soil-derived brGDGTs (BIT<0.3), which is probably caused by the too large distance to shore and/or limited soil development and subsequent transport from the land. In addition, a substantial portion of the samples analyzed has #rings-tetra values higher than 0.4 (see Table S1), which indicates that temperature estimates for these samples are likely affected by a contribution of *in situ* produced brGDGTs (Sinninghe Damsté, 2016, GCA). All of the above makes it difficult to infer a reliable long-term atmospheric temperature trend at this stage.

ORIGINAL COMMENT:
6) Calibrations

This study uses the linear calibration of Kim et al. (2010). However, it is important to note that Kim et al. (2010) plots SST on the y-axis (see Kim et al. 2010; Fig.5). As such, this calibration will suffer from a regression dissolution bias and should not be used.

REPLY:
Referee #1 is correct in stating that the calibration of Kim et al. (2010) suffers from a regression dilution bias caused by the uncertainty in the measured TEX86 values plotted on the x-axis, and we shall add this potential complication in the methods section. This bias causes flattening of the slope (Hutcheon et al. 2010, BMJ) and therefore affects TEX86-based temperature reconstructions at the lower and upper end of the calibration range. However, because TEX86 in our record lies around the middle of the total TEX86 range used for the linear calibration of Kim et al. (2010), reconstructed temperature values for Site U1356 will not be severely biased.

Indeed, the BAYSPAR method is the only TEX86 calibration that is not affected by the regression dilution. However, by applying the BAYSPAR method, only core tops within a selection of 20°x20° grid boxes are used, thereby excluding all of the low-temperature Southern Ocean core-top calibration values (Fig. 3). Considering that Site U1356A is a high-latitude Southern Ocean site, we did not want to only use BAYSPAR and, therefore, plotted the linear calibration of Kim et al. (2010) for comparison. Although this linear calibration shows large scatter at the lower temperature end, it has been shown that all Southern Ocean core-top values in the Kim et al. (2010) dataset fall within the standard error (±5.2°C) of the linear calibration of Kim et al. (2010) (Ho et al. 2014, GCA). In addition, the fact that the temperature reconstruction based on the Kim et al. (2010) calibration compares very well with the BAYSPAR-based temperature record (Fig. 3) indicates that the effect of regression dilution bias is, in our case, relatively minimal.

In the revised manuscript we will clarify our choice to use both the BAYSPAR and the Kim et al. (2010) linear calibration.

ORIGINAL COMMENT:
This paper also flips between different calibrations and there needs to be some consistency in the text and figures. For example, in Figure 3 you calculate SSTs at IODP 1356 with TEX86 (Kim 2010) and BAYSPAR (T&T 2015). However, ODP Site 511 is only shown using TEX86 (Kim 2010). Why not also recalculate with BAYSPAR?

REPLY:

We agree that there are some inconsistencies in the use of calibrations in the text and figures. We shall recalculate the temperatures of ODP Site 511 using BAYSPAR and add them to Figure 3.

ORIGINAL COMMENT:

Similarly, you only use only BAYSPAR in Figure 4, despite the manuscript stating that Kim et al. (2010) was the preferred calibration (line 241).

REPLY:

To clarify, we do not state that we prefer the linear calibration of Kim et al. (2010) over the BAYSPAR calibration. This is why we also apply the BAYSPAR calibration. The offset between these two calibrations is, however, only 0.5°C, well within the calibration error of both regressions. Because of the small temperature difference between the two calibrations it is only for convenience that we showed only one calibration in Figure 4. In the revised manuscript we will plot both reconstructions in Figure 4.

ORIGINAL COMMENT:
Minor comments:

There are a few other TEX86 datasets which might be of interest to the authors;

a. Wade et al, (2012) -> TEX86 values span the earliest Oligocene (ca. 34-32 Ma). b. Zhang et al., (2011) -> TEX86 values span most of the Oligocene.

REPLY:

TEX86 data of Wade et al. (2012), and Zhang et al. (2013, not 2011) clearly show that the early Oligocene did experience a latitudinal temperature gradient. Although a valid observation in itself, these low-latitude regions are not the focus of our study. Our study focusses on near-Antarctic SST estimates and trends and how these relate to our current knowledge of the Oligocene Antarctic ice-sheet dynamics.

ORIGINAL COMMENT:
66: state CO2 estimates for the Oligocene here

REPLY:
We will state CO2 estimates for the Oligocene

ORIGINAL COMMENT:
93: specifically, ISOPRENOIDAL glycerol dialkyl glycerol tetraethers

REPLY:
We will insert 'isoprenoidal'

ORIGINAL COMMENT:
94: Careful of the wording here. If you have GDGTs, you will likely have other fossil organics preserved. Do you mean that these are the only fossil "paleothermometers" which are preserved?

REPLY:

We mean specifically fossil paleothermometers.

REPLY:

We agree with the reviewer that this should not be part of the introduction. This section will be transferred to either the methods section 2.6 on TEX86 calibrations or the discussion of the revised manuscript.

REPLY:

We agree that BAYSPAR is not strictly a local/regional calibration. By selecting only TEX86 core-top values similar to the ones measured in Oligocene samples at Site U1356 BAYSPAR constructs a calibration based on multiple modern-day analogue sites and therefore is not regional. We shall therefore be more careful using the word 'regional'.

REPLY:

A space will be added

REPLY:

Paleoceanographic setting would be a better heading.

REPLY:

Indeed, isoGDGTs are major components in arid/alkaline soils when compared to brGDGTs. However, compared to in marine sediments, the concentration of isoGDGTs in soils is always low. We shall therefore not alter this statement.

REPLY:

This information is not explicitly given in the manuscript and we shall include this in the revised version. To answer the question, #rings-tetra can discriminate between marine and soil-derived branched GDGTs as the composition of soil-derived GDGTs typically show high amounts of the acyclic tetramethylated GDGT-Ia, while a dominance of cyclic tetramethylated (Ib, Ic) (and also pentamethylated) brGDGTs has been attributed to *in situ* production within the sediments (see Sinninghe Damsté 2016).

REPLY:

Agreed. A less detailed discussion is sufficient by stating earlier that TEX86L does not work due to its sensitivity to changes in the GDGT2/3 ratio as shown by Hernandez-Sanchez et al. (2014).

REPLY:

We shall change the sentence to "In marine sediments these lipids are assumed to originate mostly from cell membranes of marine Thaumarchaeota, because they are one of the dominant prokaryotes in today's ocean and occur throughout the entire water column".

REPLY:

Today the water depth of Site U1356 is 3992 m (see material & methods section). We have no quantitative constraints on water depth during the Oligocene, but the sediments, characterized by hemipelagic deposits reworked by bottom currents and distal gravity flows, as well as biota suggest a deep-water setting in the Oligocene (Escutia et al., 2014 Develop. Mar. Geol.). It is therefore certainly not a shallow (<1000 m depth) site. Regardless, it is more likely that GDGTs derived from shallower waters (<1000 m depth) are more effectively transported to the sediments through a.o. fecal pelleting (Schouten et al., 2013 and references cited therein). If deep archaeal communities would have contributed to the sedimentary GDGT pool, this would result in a higher GDGT2/3 ratio for these samples, which will have only a minor effect on the TEX86 (i.e. TEX86H) index (Hernández-Sánchez et al. 2014).

REPLY:

In line 280 proxies for identifying methanotrophs have mistakenly been lumped together with those identifying methanogens. We shall correct for this.

REPLY:

We will elaborate more on 'non-temperature related influences on TEX86', such as archaeal growth rates or an input from methanogenic Euryarchaeota (Zhang et al., 2016 Paleoceanography), in the revised manuscript.

REPLY:

We meant to say that in addition to the eight samples with high BIT values, another 5 were discarded based on high GDGT-0/Cren or Methane Index values. As all samples with too high BIT values also had high GDGT-0/Cren or Methane Index values, a total of 13 samples showed indications of input from methanogens or methanotrophs. We will clarify this in the revised version.

REPLY:

We shall mention the additional 155 sites aside from the Kim et al. (2010) calibration set.

REPLY:

Referee #1 is correct in stating that the BAYSPAR program gives you 90% confidence levels for each sample. Assuming that the error is normally distributed around the mean, the lower and upper 90% confidence interval boundary can be calculated as the mean plus or minus 1.645 times the standard error. As for each sample the mean, upper and lower confidence interval are known, the standard error can be calculated. On average the calculated standard error for all of the samples is ±4.2°C. The ±3.5°C standard error was miscalculated based on BAYSPAR giving the 95% confidence levels. We shall make this calculation in the methods section and correct this error.

ORIGINAL COMMENT:
331: What was the paleolatitude of the Wilkes Land section during the Oligocene?

REPLY:
We shall add the paleolatitude of Site U1356 to the manuscript, which is 58.86°S.

ORIGINAL COMMENT:
Line 404: how exactly does this resampling work? It is not clear from the text.

REPLY:
Resampling is done by predicting the $\delta18O$ value from a LOESS fit through the glacial and interglacial $\delta18O$ values. We shall explain this in the text.

ORIGINAL COMMENT:
Figure 5: Most of the data in Figure 5 is already in Figure 4. As such, I would recommend removing it.

REPLY:
We removed this figure.

**Reply to anonymous Referee #2**

Firstly, we would like to thank Referee #2 for his/her thorough feedback on the manuscript.

We recognize that the major concerns of R2 are the same as those of R1, for which we have written a more detailed reply letter.

ORIGINAL COMMENT:
In this paper Hartman et al., present Southern Ocean TEX86-derived temperature estimates from the Oligocene. They then use the TEX86-derived temperature reconstruction to de-convolve a δ18O of benthic foraminifera from an equatorial Pacific site into the δ18O of seawater component to infer relative stability of the Antarctic ice sheet during the Oligocene. I think the TEX86 record is a useful contribution to our understanding of temperature conditions around Antarctica during the Oligocene. However, I found the "thought experiment" inferring Antarctic ice volume stability (lines 419-420), particularly given the limitations of TEX86, unconvincing. The authors are very mindful about the limitations of this "thought experiment" but inferring ice volume stability feels over-reaching. I think a revised manuscript should focus on the regional oceanography and polar frontal systems. I also found it difficult to comprehensively review this manuscript because it references two submitted articles Salabarnada et al., and Bijl et al.,) related to the lithology and surface water conditions.

REPLY:
Similarly to our reply to R1, we agree with R2 and will focus on the relation between temperature and δ18O in a more qualitative way, and redirect our discussion more towards the nature of the high variability in the SSTs in a revised version of our manuscript. In addition, we agree with Referee #2 that the discussion is too much focused on the one scenario that links the TEX86 variability to the δ18O variability, while the discussion on the role of oceanography (polar front shifts) and its link to temperature changes is limited. Indeed, the role of oceanography is also very important for the stability of the cryosphere (Sangiorgi et al. 2018, Nature Comm.) and it is therefore highly interesting to analyze our temperature data in light of possible oceanographic changes. In the revised manuscript we will therefore explore scenarios that involve the potential role of shifting polar frontal systems over Site U1356 to explain the TEX86 variability.

We are sorry that Referee #2 felt unable to comprehensively review this manuscript because it refers to two submitted papers (Salabarnada et al. and Bijl et al.), both in review in CP. By submitting the three manuscripts back-to-back in Copernicus journals, we hoped to enable all reviewers to openly access them for the purpose of their own review. We understand that submitting 3 back-to-back papers implies that the reviewers should read and evaluate all three papers to comment on one of the them. Salabarnada et al. submitted their rebuttals at at https://www.clim-past-discuss.net/cp-2017-152/#discussion and the reviews for Bijl et al. can be found here (https://www.clim-past-discuss.net/cp-2017-148/#discussion). We believe that none of these reviews suggest a major reconsideration of our conclusions.

ORIGINAL COMMENT:
There are many fundamental assumptions the authors make and explore within the manuscript with

respect to relative contribution of temperature and ice volume in the δ18O of benthic foraminifera from Site 1218:

1) Temperature bias in TEX86 is to summer and deep water production (in the modern) is to winter. The authors note this throughout the manuscript but this is a difficult temperature disconnect to constrain. What is the seasonal range in temperatures from summer to winter today? The winter temperatures during deep water formation are constrained because seawater freezes at -2C. Summer temperatures, particularly in a warmer world, could vary by a lot. In particular, the TEX86-derived temperatures are nearly twice that of the Mg/Ca bottom water temperature record.

REPLY:
We agree with the reviewer that this temperature disconnect is difficult to quantify for the Oligocene, based on our data. In general, quantification of temperature disconnect for past periods before monitoring became available is always difficult and has at best to rely on proxies (and their uncertainties). Because of this, we do agree we should refrain from quantifying the temperature signal in δ18O using our TEX86 results. However, we would like to point out that the temperature at the locus of deep-water formation could have changed profoundly in the past. Deep-sea δ18O changed considerably over the ice-free Eocene (Zachos et al., 2008), which can only result from changes in deep-water temperature in the absence of ice sheets. Although on Oligocene glacial interglacial time scales we expect SST changes to be in part affected by the migration of polar frontal systems, we cannot rule out that winter temperatures at the locus of deep water formation changed as well. After all, the locus of deep water formation is to some extent predefined by geographic boundaries, next to the physical properties of the water that cause the water to sink. Therefore, the possibility that a significant part of the benthic foraminifer δ18O variation of Site 1218 is related to glacial-interglacial winter temperature variability remains. We shall focus the discussion more towards this point in a revised version of our manuscript.

ORIGINAL COMMENT:
2) No subsurface temperature bias in TEX86. Given that the temperatures vary by >10C and this assumption relies on a submitted manuscript (Salabarnada et al.,), I found this assumption difficult to evaluate. My main concern is that "interglacial" and "glacial" temperatures are related to lithology. The packaging and flux of TEX86 to the deep ocean is likely very different during these times: "interglacial" temperatures are during bioturbated carbonate-rich periods and "glacial" are during laminated silty periods. Also, are there post-depositional processes that might influence TEX86 estimates due to the change in lithology?

It seems given the uncertainty and variability in the temperature reconstructions, all calibrations should be discussed, including the subsurface ones. The BAYSPAR calibration itself (Tierney and Tingley, 2015), which is discussed in some detail, uses regional factors such as the vertical temperature gradient and related subsurface TEX86 influence to reconstruct temperature.

REPLY:
The sedimentation of Site U1356 during both glacial and interglacial periods is characteristic for a deepwater distal setting, dominated by fine-grained turbidite overbank and hemipelagic depositions, that are reworked by bottom currents of different intensities (stronger during the interglacials). There is therefore lithologic variation between glacial and interglacial time periods, but we believe the processes responsible for the lithologic variability are not the ones that typically change GDGTs. Post-depositional processes, such as oxic degradation, do not affect the ratio between the various isoGDGTs (Huguet et al. 2009, OG). To investigate if there is any relation between lithology and archaeal community changes, we checked if there is any correlation between lithology and GDGT2/3 ratio, which can identify potential contributions of subsurface GDGTs. However, no relation between the lithological facies and the GDGT2/3 ratio has been found. We agree that this should be more adequately discussed, and will change this in the revised manuscript.

As Referee #2 mentions, there is significant variability in the temperature reconstructions across the various calibrations to (sub)surface temperature (Figure S2). However, in our materials & methods section we thoroughly discuss why most of these calibrations cannot be applied at Site U1356.

ORIGINAL COMMENT:
3) The authors mostly dismiss the Lear et al., 2004 Mg/Ca bottom water temperature record. This is odd because the Mg/Ca record is from the same site as the δ18O of benthic foraminifera (Site 1218) so it should be discussed in some length. The authors note changes in Mg/Ca of seawater and carbonate ion may influence the Mg/Ca-based temperature reconstruction. There are many uncertainties about the Mg/Ca paleotemperature sensitivity to changes in Mg/Ca of seawater (Evans et al., 2016 and for a nice discussion see Lear et al., 2015 for ice volume estimates for the Miocene to present) but the relative direction in bottom water temperatures shouldn't be an issue. The fact that the TEX86 and Mg/Ca-derived temperature estimates have different trends can't be explained by Mg/Ca of seawater changes. Additionally, the benthic foraminifera used in the Lear et al., 2004 study are an infaunal species, largely insulated from longterm changes in in carbonate ion (Lear et al., 2015, Ford et al., 2016).

REPLY:
We agree with Referee #2 that we too easily dismissed the Lear et al. (2004) Mg/Ca bottom water temperature record. We thank Referee #2 for pointing out to us that the benthic foraminifera *Oridorsalis umbonatus* used for reconstructing the Mg/Ca record of Site 1218 is infaunal and therefore to some extent, but not completely insulated from long-term changes in carbonate ion concentrations, and also for citing Lear et al. (2015), which states that this species is insensitive to changes in the Mg/Ca ratio of the seawater. We shall revise the discussion of the Lear et al. (2004) benthic Mg/Ca record in the revised manuscript. Instead of focusing mostly on the relation between our TEX86 record and the benthic δ18O record of Site 1218, we shall discuss several scenarios that will try to explain the differences and similarities between the long-term trends and the variability of our TEX86-based surface water temperature record, the Mg/Ca-based bottom-water temperature record of Lear et al. (2004), and the δ18O record more extensively.

ORIGINAL COMMENT:
4) Given the changes in lithology and the offset between the glacial and interglacial LOESS curves is constant, I'm not sure resampling the "glacial (values above average _18O) and interglacial (values

below average δ18O) δ18O trends at Site 1218" (lines 403-405 is the best approach. In fact, I think much of the discussion in the section "4.3 sea surface temperature variability at glacial and interglacial time scales" is poorly supported given the uncertainty in the age model, lithology, and TEX86-based temperature estimates. A more thoughtful approach would a comparison figure of mean δ18O of seawater estimates from 1) LOESS TEX86 and a LOESS δ18O of benthic foraminifera and 2) the high-resolution Mg/Ca and δ18O of benthic foraminifera.

REPLY:
Alternations between laminated and bioturbated carbonate-rich sediments allow us to identify orbital glacial-interglacial cyclicity. Although we cannot identify each orbital cycle within our record due to core gaps, the age model of U1356 is definitely robust enough for reconstructing long-term trends in the SST reconstructions. If we would not distinguish between TEX86-derived temperatures from glacial lithologies and those obtained from the interglacial lithologies, the temperature trend would flatten out due to the larger internal temperature variation. In addition, we would like to remind R2 that our TEX86 record is of relatively low resolution, and sample location is guided (in part) by avoiding sediments that are known not to be in situ (i.e., distal mass transport deposits). For this reason, samples are in places predominantly obtained from glacial sediments, while in other places they are predominantly obtained from interglacial sediments. Running a LOESS curve through the entire dataset could therefore potentially establish non-existing trends due to our irregularly-spaced sample distribution. In order to resolve this uneven distribution of samples from the two different lithologies, we apply a LOESS curve on the (independently separated) glacial and interglacial data(sub)sets. We are confident that this better reflects the actual temperature trends, because both the glacial and interglacial LOESS curve show the same long-term trend despite the fact that they are based on separate data(sub)sets. ｌSince it would be an uneven comparison to compare these to the mean of δ18O, we have chosen to compare glacial SSTs to the above-average δ18O and the interglacial to the below-average δ18O.

Secondly, Referee #2 suggests comparing δ18O of the seawater calculated by using TEX86-based temperatures versus δ18O of the seawater calculated by using Mg/Ca-based temperatures. Upon reviewing our initial approach, and supported by the comments of both R1 and R2, we will no longer quantify δ18Osw changes from our TEX86 record. Since we will discuss matches and mismatches between the trends of the δ18O record and the TEX86-based temperature record more qualitatively in the revised manuscript, resampling of the benthic δ18O record is no longer valid.

ORIGINAL COMMENT:
5) The Site 1218 δ18O of benthic foraminifera is used because it covers the entire record. However, are the trends in δ18O of seawater different when the other high resolution Site 1264 δ18O of benthic foraminifera is used? Any one location can be influenced by changes in hydrography.

REPLY:
The long-term (million-year) trend of Site 1264 is very similar to that of Site 1218 (Liebrand et al. 2017, PNAS) and there is therefore no difference in the reconstructed δ18O of the seawater. In fact, globally all Oligocene δ18O records follow the same trend, except for the δ18O record of Maud Rise (Hauptvogel et al. 2017, Paleoceanography), which we believe is indeed influenced by changes in hydrography.

ORIGINAL COMMENT:

Minor comments: The authors should include changes in paleolatitude and whether that might influence the temperature record. Are there large changes in sedimentation rate that might influence preservation and/or these records?

REPLY:
We shall include the paleolatitudes of Site U1356. Site U1356 shifted from 58.86°S at 30 Ma to 59.43°S at 22 Ma (using van Hinsbergen et al. 2015, PloS One). We believe that this shift to higher latitudes could be at least partly responsible for the increased glacial-interglacial temperature variability in the late Oligocene. We acknowledge that this is not part of the manuscript in its current state and we will include this discussion in the revised manuscript in the part that focusses more on the potential role of Southern Ocean fronts.

Changes in sedimentation rate in general will not affect the temperature reconstruction, as the TEX86 (i.e. the relative abundance of GDGTs) relatively unaffected by oxic degradation (Kim et al., 2009, GCA) and if so, this would result in substantially elevated BIT indices something we do not observe (Huguet et al. 2009, OG). Changes in sedimentation rate at Site U1356 are mainly determined by the deposition of mass transport deposits. These could contain allochthonous GDGTs, which is why samples from this type of deposits were not used for reconstructing the sea surface temperature record.

**Response to Stephen Gallagher (referee)**

We thank Stephen Gallagher for reviewing our manuscript and for acknowledging the value of our dataset. His annotations to our manuscript showed us that some sections lack clarity. In particular the last part of section 4.2, which involves the "thought experiment", and section 4.3 that discusses the reconstructed temperature variability. Also, considering the comments of Referees #1 and #2, we have decided to significantly restructure those sections, placing more emphasis on the role of paleoceanography (polar fronts), and to refrain from quantifying ice volume changes. In the revised manuscript we will discuss several scenarios that can explain the differences and similarities between our TEX86-based temperature record, the benthic δ18O records and the Mg/Ca-based bottom-water temperature record. We believe that this approach will improve the structure and clarity of the manuscript.

ORIGINAL COMMENT:
This is very good new organic proxy dataset from offshore Wilkes Land. The authors present a near field palaeotemperature record that although is much lower in resolution compared to other proxy datasets it sheds light for the first time consideration of the long term sea surface temperature evolution of this Wilkes Land margin.

I have made extensive comments and suggestions in the attached annotated text to this paper.

REPLY:
All comments and suggestions in the annotated text were clear, mostly they were related to the choice of words or to incorrect English and suggestions will be included in a revised manuscript. The reviewer's major comments are addressed below.

Comment: In the annotated manuscript, in line 238, the reviewer asks for clarification on why TEX86 would not have been influenced by subsurface temperatures in the absence of low-salinity waters due to sea ice melt.

Answer: It has been shown that the sea-ice influenced, low-salinity surface waters of today's Southern Ocean contain virtually no Thaumarchaeota in the top layer of the water column (0-45 meter below sea level (mbsl), Kalanetra et al. 2009 Environ. Microbiol.). Instead, the GDGTs are derived from Thaumarchaeota living in the deeper water column (45-105 mbsl, Kalanetra et al., 2009 Environ. Microbiol.) and, therefore, TEX86 does not represent a true surface water signal (see also Kim et al., 2012 Geophys. Res. Let.). Dinoflagellate cysts in the same sediments suggest that oceanographic conditions were similar to today's Subtropical Front and that no sea ice was present. Hence, we conclude that TEX86 at Site U1356 does reflect a surface water temperature. We will clarify this in a new version.

Comment: In line 251 Stephen Gallagher has placed a question mark at the word 'prior' in the sentence "The prior for site U1356 is obtained from recent clumped isotope measurements (Δ47) on planktonic foraminifers from Maud Rise (ODP Site 689) (Petersen and Schrag, 2015), which show early Oligocene temperatures of 12°C."

Answer: the BAYSPAR method is based on Bayesian inference and therefore requires a prior distribution of temperature (i.e. the prior) in order to predict sea surface temperatures from the observed TEX86 values. In general, the prior is our initial belief or scientific understanding of the unknown quantities to be estimated, in this case sea surface temperature (Tierney & Tingley, 2014). As for deep time temperature reconstruction this prior cannot be based on modern-day annual mean sea surface temperatures, the BAYSPAR method requires a user-specified mean and variance for this prior (Tierney & Tingley, 2014). Therefore, we use previous estimates of southern high-latitude early Oligocene seawater temperatures (12°C based on Δ47) as a prior for our TEX86-based sea surface temperature reconstruction. We will better explain this in the revised manuscript.

ORIGINAL COMMENT:
I would like to add the following to the discussion: Reference to EAIS volume changes in line 70 page 3: As I iterated in my review of the Bjil et al submission to this volume: I appreciate the utility of using isotopes to interpret Antarctic Ice Sheet variability as summarise by Liebrand et al (2017) (www.pnas.org/cgi/doi/10.1073/pnas.1615440114) and this approach is used extensively when discussing the Cenozoic greenhouse icehouse transition. However, there are other sections that have been interpreted using backstripping and stratigraphic data in the Gippsland and New Jersey margins that reflect glacio-eustasy in the Oligocene and relative ice volume (Gallagher et al., 2013), it would be useful to consider the significance of these near field and far field sections in any section reviewing ice volume variability. This paper also considers the apparent instability of the EAIS during the Oligocene and presents a sea level curve with Oi events (Figure 6 in Gallagher et al; at slightly higher resolution that the present study) that bears striking similarity to the temperature curve presented in this paper (Figure 4 in this submission).

Gallagher, S. J., G. Villa, R. N. Drysdale, B. S. Wade, H. Scher, Q. Li, M. W. Wallace, and G. R. Holdgate (2013), A near-field sea level record of East Antarctic Ice Sheet instability from 32 to 27 Myr, Paleoceanography, 28, doi: 10.1029/2012PA002326.

REPLY:
We agree that the sea level curve reconstructed in this paper shows the same long-term trends as our temperature record. In our new version of the manuscript, we will sketch several scenarios to explain the differences and similarities between our TEX86 record and the benthic δ18O record in a more qualitative way. As this will likely involve ice volume changes and therefore global sea level changes, the paper by Gallagher et al. (2013) will be a nice addition to this discussion, providing a framework for our theories.

ORIGINAL COMMENT:
More specific comments are below:

Line 95: The core recovery in the Wilkes Land section is certainly not "complete"

REPLY:

The reviewer is correct in this and we will revise the sentences where we give the impression that Hole U1356A is without hiatuses.

ORIGINAL COMMENT:
Line 145 page 5: These are modelled plate tectonic reconstructions.

REPLY:

We will correct this. Indeed tectonic reconstructions show that Australia and South America were closer to Antarctica. We meant to say that numerical modeling of ocean currents shows that the strength of the circum-polar current was limited by these narrow gateways (Hill et al. 2013).

ORIGINAL COMMENT:
Line 160 reference to Bijl et al paper in Jl Micro to be cited?

REPLY:

We will cite Bijl et al. (2018, J. Micropal.) at the end of this line to refer to the position of *M. escutiana*.

ORIGINAL COMMENT:
Line 315 I agree to a certain extent about the lack of identification of Oi events due the gaps in the record (not unexpected during glacials near Antarctica), however, Oi2 is not near 32 Ma (Figure 4) it is actually near 30 Ma and it is possible there is core of this event in the section (see possible correction of Figure 4).

REPLY:

We will correct for this mistake and place Oi-2 near 30 Ma in Figure 4. Although it might be possible that Oi-2 is recorded in U1356A, age model limitations prevent us to be certain.

ORIGINAL COMMENT
Pages 13 and 14: This section is very interesting yet requires significant clarification, I have suggested ways to enhance the message and tone down the "speculation" in this section hopefully these suggestions help.

REPLY:

We will reduce the speculation by refraining from a quantitative comparison between TEX86 and benthic δ18O.

ORIGINAL COMMENT:
In conclusion, once the text has been clarified and the suggestions considered this will be another useful addition to the relatively sparsely documented Antarctic (palaeo)climate and oceanographic records.

REPLY:

We thank the reviewer for these positive comments.

**List of all relevant changes made:**

- The title was changed to include the Miocene data of Sangiorgi et al. (2018) (see below).
- Henk Brinkhuis has been added as a co-author to this manuscript, due to his participation as co-chief to IODP Expedition 318 where the material comes from and his valuable contribution to this new version of the manuscript.
- Because of the movement of the Earth Sciences department of Utrecht University, the address was changed.
- The corresponding email address has been changed.
- Section 2.6, which discusses the $TEX_{86}$ calibrations has been rewritten for a large part in order to shorten the discussion on $TEX_{86}^{L}$, as proposed by Referee #1, and to improve our argumentation for the use of the BAYSPAR calibration and the linear calibration of Kim et al. (2010). Both these calibrations have also been consistently applied to all $TEX_{86}$ data (incl. published data) and plotted in all figures, as was suggested by Referee #1.
- GDGT-2/GDGT-3 ratios have been added to Figure 2 and are discussed in the text. A correlation of the GDGT-2/GDGT-3 ratios to lithology, which may have implications for the interpretation of our dataset as Referee #1 and Referee #2 suggested, has now been excluded.
- Data of the mid-Miocene section published by Sangiorgi et al. (2018, Nat. Comm.) has been added to the record for consistency between the three different papers that are jointly submitted. The same analytical procedures as for the Oligocene data have been applied to this data. Of particular importance is the relation of reconstructed Miocene SSTs to the lithology, which has not been discussed by Sangiorgi et al. (2018). The lithology for the Miocene (based on Salabarnada et al., submitted this volume) has been added to Figures 2 and 4.
- For comparison, Miocene sea surface temperature estimates from other Sites (ODP Site 1171 and ANDRILL-2A) have been added, as well as bottom-water temperature estimates based on Mg/Ca records from ODP Site 1171 and ODP Site 747.
- In the previous manuscript only the high-resolution benthic $\delta^{18}O$ records of Site 1218 and Site 1264 were included in the discussion as well as Figure 4. To extend the benthic $\delta^{18}O$ record into the Miocene and to show that the records of Site 1218 and 1264 follow the global benthic $\delta^{18}O$ trend (questioned by Referee #2) other records have now been included and normalized to Site 1264 to get a good representation of the global average trend (LOESS curve). The benthic $\delta^{18}O$ record is not resampled as in the previous manuscript as a quantitative approach for the comparison between reconstructed SST long-term trends and benthic $\delta^{18}O$ long-term trends has been abandoned.
- The discussion is thoroughly revised based on the suggestions of the reviewers. In the discussion our reconstructed sea surface temperature record is compared to the $\delta^{18}O$ record in a more qualitative way. Notably, the so-called thought experiment that deduced ice volume from based on our sea surface temperature record was considered too far-fetched by all reviewers, and has therefore been removed. Several scenarios are discusses to explain the differences and similarities between trends observed in the sea surface temperature, $\delta^{18}O$ and Mg/Ca-based bottom water temperature records, instead of focusing on the transport of a sea surface temperature signal to the deeper waters only. Included in this discussion are now also

paleogeographic changes, shifts of Antarctic frontal zones (as proposed by Referee #2) and the effect of the Antarctic ice sheet on sea surface temperature.

- Figure 5 was removed, as suggested by Referee #1.

[revised manuscript text omitted]

---

## Author Response (AR2)

**Universiteit Utrecht**

**Editorial Board, Climate of the Past**

To the editorial Board

**Faculty of Geosciences,**
**Department of Earth Sciences**
Marine Palynology and
Paleoceanography

**Visitors Address**
Princetonlaan 8a
CB Utrecht
The Netherlands

**Your reference**
**Our reference**   Climate of the Past manuscript
**Phone**           +31 30 253 2630
**Email**           J.D.Hartman@uu.nl
**Website**         https://www.uu.nl/staff/JDHartman
**Date**            August 8, 2018
**Subject**         Submission of manuscript to Climate of the Past (cp-2017-153)

Dear dr. David Thornalley,

We apologize for the delay in this response due to summer break. Thank you for the week additional time allowed for the final resubmission of this manuscript.

Please find enclosed a final and marked-up version of our manuscript, entitled "**Paleoceanography and ice sheet variability offshore Wilkes Land, Antarctica – Part 3: Insights from Oligocene-Miocene TEX$_{86}$-based sea surface temperature reconstructions**", as well as a point-by-point reply to our reviewers. Note that we have changed the title in accordance with the already published companion papers by Salabarnada et al. (2018) and Bijl et al. (2018).

We thank both reviewers for their positive comments on this new version. Based on the input by the reviewers, the most significant changes were made in Section 4.3 that involves glacial-interglacial temperature variability. As suggested by Reviewer #1, we have added some comments here on the temperature variability within a single sedimentary facies. Another concern of Reviewer #1 was that there is the possibility that reconstructed temperatures reflect vertical shifts in the position of Thaumarchaeotal GDGT export within the water column instead of true surface water temperature changes. We have considered his/her comment carefully and following his/her suggestion we have added a part of the discussion to demonstrate why we think that there is a very small chance that a migrating depth habitat of the Thaumarchaeota has affected our TEX$_{86}$-based SST reconstructions (see also the detailed reply to the reviewer).
The concern of Reviewer #2 was related to the fact that we use a summer-biased proxy to qualify the temperature contribution to the benthic δ$^{18}$O stack. Reviewer #2 suggested that in order to support this theory we could compare the magnitude of change between our reconstructed SST record and the Mg/Ca-based bottom water temperature records for both glacial and interglacial datasets. In the detailed reply to Reviewer #2 we explain why this suggested exercise is in our view risky as it entails a series of assumptions on the Oligocene oceanography of the Southern Ocean we have no constraint for.

Figures were changed according to the suggestions by the reviewers. Following the suggestion of Reviewer #1, the Kim et al. (2010)-based data and LOESS curves have been removed from Figure 4, which makes it easier to read.

The manuscript contains 4 figures, 2 supplementary figures and 2 supplementary tables and about 10,000 words.

In anticipation of your reply,

Best regards, also on behalf of the co-authors,

Julian D. Hartman
Corresponding author

**Reply to Referee #1**

Original comment:
Firstly, thanks to the authors who have engaged with many of my comments. Overall, I think that the dataset presented here will be a useful addition to the Cenozoic paleoclimate community. However, it still remains unclear to me why TEX86 values should fluctuate so widely during the Oligocene (i.e. from 10 and 21'C).
I have explored some of these comments in detail below:
1) Subsurface production over glacial/interglacial timescales
I am glad to see that both companion papers (Bijl et al. and Salabarnada et al) have now been accepted and/or published (congratulations!). As such, I have more confidence in the authors interpretation re: glacial/interglacial cycles.
However, I would like to point the authors towards an elegant paper published in 2016 (Hertzberg et al; EPSL). Here, the authors reconstruct TEX86 SSTs during the last glacial and interglacial cycle. These estimates are then compared alongside Mg/Ca-derived SST estimates from: 1) upper mixed layer- (0 to 50m), 2) upper thermocline- (25 to 85m) and 3) lower thermocline (60 to 150m)-dwelling foraminifera. During interglacials, TEX86 SSTs closely match Mg/Ca values from mixed layer dwellers. However, during glacials, TEX86 values are much colder (up to 5°C) and appear to represent thermocline temperatures. In other words, there is deeper export of GDGTs during glacial periods.
Therefore, is the glacial/interglacial TEX86 variation at Wilkes Land reflecting a true temperature signal? Or is it just migration of thaumarchaeota in the water column and deeper export?

Response:
We thank R1 for his/her positive words and final comments, including the suggestion of considering our data in view of the publication by Hertzberg et al. (2016). Indeed, we cannot completely rule out the possibility that the temperature signal comes from different water depths (where thaumarchaeota are produced) during glacial vs. interglacial intervals. Yet, we have added a discussion on this aspect in paragraph 4.3 and discuss why we are still confident in our data (lines 641 and following). We point out that the explanation for a migration of highest GDGT export production towards lower depths given by Hertzberg et al. (2016) involves a lowering of the nitrite maximum in the water column in association with a decrease in nutrient availability and productivity. At one of their sites this is also supported by an increase in the GDGT-2/GDGT-3 ratio, which increases with depth. At Site U1356 we find no relation between the GDGT-2/GDGT-ratio and the glacial and interglacial sedimentary facies (Fig. 2). Also, Site U1356 is characterized by more oligotrophic conditions during the interglacials, which would mean that, following the reasoning of Hertzberg et al. (2016), interglacial SST estimates reflect a subsurface temperature signal. If this is the case, our reconstructed temperature variability would even underestimates true temperature variability during the Oligocene. We added this discussion to the manuscript as well, in our exploration of the potential bias of depth of production of GDGTs.

Original comment:
2) TEX86 variability

There still seems to be large variability in TEX86 values which are unaccounted for by the authors. For example, there can be ca. 5 degrees of variation in the glacial deposits over a very short time interval (e.g. 31 Ma). In your reply, you state that you will discuss this in the manuscript; however, I could not find any reference to this. Could you please point me towards the correct section?
Also, 5 degrees of warming is not insubstantial. The Paleocene-Eocene Thermal Maximum is characterised by a similar magnitude of surface warming (e.g. Sluijs et al., 2006).

Response:
R1 is correct in stating that this is not discussed. We have added a short paragraph on the temperature variability within a sedimentary facies within section 4.3. Here we point out that the glacial-interglacial variability that we reconstruct based on the sedimentary facies reflects obliquity-paced variability. SST variability within one sedimentary facies can therefore reflect variability caused by precession-paced changes in insolation.

Original comment:
Minor comments:
The authors state that they are unable to conduct a data-model comparison. However, there is a wealth of literature which they could draw on (e.g. Sijp et al., 2014; Global and Planetary Change).

Response:
R1 is correct in stating that several studies reconstructed past oceanographic conditions using numerical models, but none of these specifically applied Oligocene boundary conditions in these models. The specific boundary conditions used are crucial for accurate model-data comparisons, notably the precise position of continents, tectonic configuration of gateways, depth of conduits, Antarctic ice sheet size and grid cell size. We find that none of the available model simulations allow for a meaningful model-data comparison to put against our Oligocene data, because these boundary conditions are very different from what we now know of the Oligocene. In Sijp et al. (2014) an Oligocene (30 Ma) plate tectonic configuration is shown, but only in relation to modeled deep water temperatures. Models with the Tasmanian Gateway closed cannot be representative for the Oligocene situation at Wilkes Land as due to this barrier eastward flowing ocean currents along the south coast of Australia (proto-Leeuwin Current) will be diverted towards the south and will affect Site U1356. However, no modeling studies exist that show the influence of the warmer proto-Leeuwin Current on Wilkes Land SSTs when the Tasmanian Gateway has opened, but not sufficiently to allow for significant proto-Antarctic Circumpolar Current throughflow (Hill et al., 2013). Also, it has been suggested that Drake Passage was closed during the late Oligocene (Lagabrielle et al., 2009). Again, a continental configuration with closed Drake Passage and an open Tasmanian Gateway (as in Hill et al., 2013) has not been used with sufficiently high spatial resolution for modeling SSTs.

Original comment:
Figure 2: The colour used to distinguish glacial and mixed facies looks indistinguishable (both look green).

Figure 2: is the key colourblind safe? Check here: http://colorbrewer2.org

Response:
The colour of the mixed facies has been changed in Figure 2 so that it is more distinguishable. Also, it has been checked if the rest of the chosen colours for the key are colourblind safe by doing a proof setup view in Adobe Illustrator. To make it more colourblind safe, the colour for the 'slump with clasts' facies has been altered in Figure 2 and 4.

Figure 3: Why do you only show error bars for just Wilkes Land?

Response:
Error bars are now also plotted for other $TEX_{86}$-based temperature estimates (i.e. AND-2A and Site 511).

Figure 3: It may be more useful to plot a LOESS regression through ALL of the Southern Ocean data to understand "Southern Ocean" temperature variability during the Oligocene and Miocene. It would also add more value to your dataset.

Response:
Plotting a LOESS curve through all the data will result in a skewed average when data are from other records and different sites, and not continuous throughout the studied time interval. We do not think that a regression through all data available would necessarily provide a better understanding of the Southern Ocean temperature variability. This exercise would definitively be useful when applied to continuous temperature records from sites along a Southern Ocean transect.

Figure 4: this figure has a LOT of information on here. Most of it is essential, but there could be room for improvement. For example:

a) The datapoints which are used to denote changes in IRD, sea-ice dinocysts etc currently overlap within the x-axis for SST.

b) "TEX86-based glacial and interglacial SSTs" has the subheading "linear calibrated SSTs". However, the corresponding "BAYSPAR SSTs" has no x-axis. Please amend.

Alternatively, just get rid of TEX86 (Kim et al.) and just show BAYSPAR. As the trends are identical (i.e. Figure 3), you only need to show one.

Also, please note that BAYSPAR is LINEAR calibrated!...it just treats uncertainty better.

Response:
We changed Figure 4 according to the suggestions made by R1, and removed the Kim et al.-based data and LOESS curves from the Figure. We agree with R1 that this makes the figure more readable. Changes have been made in the manuscript text to account for the Kim et al.-based LOESS curves not being plotted.

c) In the Miocene, what are the straight lines in the "linear" and "BAYSPAR" TEX86 SST plots? This really confused me.

Response:
This is clarified in the text, but was lacking in the figure caption. We have now clarified this in the figure caption.

Original comment:
Line 240: how do you define "too low" when referring to concentrations of GDGTs?

Response:
"too low" has now been specified in the manuscript: "peak height less than 3x background, as well as peak areas below $5*10^3$ mV and $3*10^3$ mV for HPLC-MS and UHPLC-MS, respectively".

Original comment:
Line 285: The latest global core-top calibration is Tierney and Tingley.

This sentence has been reformulated and no longer includes "The latest global core-top calibration".

Original comment:
Line 355: you appear to have said 132 samples earlier.
Response:
R1 refers to line 221 for the 132 samples, while 113 samples are mentioned in line 355. However, both statements are correct. The 132 samples still include the 16 samples that were discarded based on too low GDGT concentrations, leaving us with 116 samples that are useful for GDGT analysis. Of these 113 samples are Oligocene samples used for GDGT analysis and 3 are earliest Miocene. These 3 other samples are now added to the 113 samples in the manuscript to be more precise. Together with the 29 samples of Sangiorgi et al. (2018) they make 145.

Original comment:
Line 388: It might be nice to show histograms of the glacial and interglacial TEX86 values…this would illustrate the offset between the two datasets.

Response:
Although we agree that a histogram would illustrate the offset between the two datasets, we believe this offset is already clearly illustrated in Figure 4 as the difference between the glacial and interglacial LOESS curves. The t-test significance values provide sufficient evidence for the difference between the glacial and interglacial datasets.

Original comment:
Line 373: they can also produce GDGT-1, 2 and 3…just in minor abundances

Response:
This is correct and also the reason why we test for the contribution of Euryarchaeota to the isoGDGTs in our samples. We have changed the text as suggested by R1 for clarification.

Original comment:
Line 463: some of your paleolatitudes are VERY precise. I am not sure we can be accurate to one decimal place!

Response:
The reviewer is correct. These are mean paleolatitudes obtained from paleolatitude.org (van Hinsbergen et al., 2015), which come with an error bar that is larger than 1°. We have changed the paleolatitudes to approximations without decimal places.

Original comment:
Line 510: I would argue that "striking" is too strong here.

Response:
"striking" has been changed into "good".

Original comment:
See also new paper by Super et al. in Geology with Miocene and latest Oligocene SSTs
Response:
We thank R1 for pointing out this paper. As is evident from this paper temperatures were between 24 and 35°C in the North Atlantic during the Late Oligocene and the Miocene Climatic Optimum. We now refer to this paper, in order to put the warmth we see in our record into better context

**Reply to Referee #2**

Original comment:
This is a re-reivew of Hartman et al.,

I think the manuscript has greatly improved. My main concern is still the interpretations made from using a summer-biased SST record to qualify the bottom water temperature contribution to a benthic d18O stack.

Today, bottom waters form during winter months and in some locations, are dependent on sea ice processes. I don't know what the formation processes were in the Oligocene or if there was much temperature contrast between summer and winter months. Presumably the annual temperature contrast was quite large if there was sea ice formation (20C during interglacials?).

One possible way to investigate the temperature variability would be to partition the Lear et al., record, which would mostly be winter time temperatures, into interglacial and glacial intervals. The authors could then compare the magnitude of temperature change during interglacial SST and bottom water temperature and glacial SST and bottom water. If the range in temperature is similar between interglacial SST and bottom water temperature and glacial SST and bottom water, I think that might help support the argument that the benthic record is mostly temperature. That is, is the range in temperature during the summer (TEX) similar to the winter (Mg/Ca)? I'm not sure how else to explore this temporal disconnect between the records.

Response:
We thank R2 for reviewing our revised manuscript and for the positive comment.
We understand the concern of the reviewer that it seems illogical to use a potentially summer-biased SST (TEX) record to qualify a winter/bottom water temperature contribution to a d18O stack. However, as the reviewer states, oceanography during the Oligocene was probably very different from today and formation of bottom water through brine rejection during sea ice formation was probably limited to the southernmost location of Antarctica (e.g., the Ross Sea). The few samples where we have sea ice dinocyst indicators in very low percentages in the (early) Oligocene may even indicate that sea ice was transported (e.g., from the continental shelf) instead of produced in situ as we write in the paper. It is possible that deep waters formed offshore the Antarctic coast through downwelling of surface waters (see Goldner et al., 2014), which is a different phenomenon than the bottom-water formation through brine rejection. However, this deep water is what influences most carbonate records from where foraminiferal isotope records are derived. If the dominant mode of deep water formation along the Antarctic coast is not through brine rejection, but occurred via surface waters downwelling, it is more logical that bottom water temperature records and benthic $\delta^{18}O$ records from far-field sites follow Southern Ocean SST trends closely. Importantly, this process is also to a much lesser extent confined to just the wintertime. We would like to emphasize that this is a hypothesis that was formulated to explain the remarkable similarity between the reconstructed SST trend and the trend of the benthic $\delta^{18}O$ stack, in the same way as for the Eocene in the region (Bijl et al., 2009). This similarity cannot be denied and is independent from our hypothesis.

For the reasons above, we do not feel comfortable in adding the results of the exercise suggested by R2 to the manuscript, because it would entail another series of assumptions. Particularly, this exercise would partly bring the paper back to the "too far-fetched exercise" that the reviewer criticized in our earlier version. As R2 points out, there is no reason to believe that the magnitude of change for winter/bottom water temperatures and summer SSTs is the same. In fact, seasonality might change dramatically in the presence of sea ice formation. Also, bottom water temperatures could be influenced by changes in the location of bottom water formation and by mixing with other water masses, especially when considering that Site 1218 lies in the equatorial Pacific and thus very far from the source region. Notably, Mg/Ca-based bottom water temperatures record a trend during the late Oligocene at Kerguelen Plateau (Site 747) that is opposite to the trend at Site 1218 (Fig. 4).

We remain confident that our reconstructed long-term SST trend reflects relative long-term temperature change in the Wilkes Land region and that a plausible explanation for its resemblance to the benthic $\delta^{18}O$ stack is the incorporation of this SST trend into the deeper waters.

Original comment:
Line 145 – omit "much more"
Line 224 – maybe use disturbed or perturbed instead of contorted
Line 357 – contorted is an odd usage
Line 376 – space after 3

Response:
We accepted the minor textual changes suggested by R2 above, We replaced "contorted", with "disturbed".

Original comment:
Figure 3A – please indicate the modern range in temperatures on the Y axis

Response:
The modern-day temperature range has now been added to Figure 3A as well as in the text (Section 2.1).

**List of all relevant changes made:**

- The title was changed so that it would link more clearly to the companion papers.
- The companion papers are explicitly referred to in the introduction.
- Because the companion papers have been published, all references to these papers have been altered.
- Modern-day sea surface temperatures are indicated in Section 2.1 and in Figure 2
- In Section 2.5 it has been specified what we consider too low concentrations of GDGTs.
- The reference to Super et al. (2018) has been included, as suggested by R1, to sketch a global picture of sea surface temperatures during the late Oligocene and Miocene (Section 4.1).
- The reconstructed SSTs and LOESS curves based on the Kim et al. (2010) linear calibration have been removed from Figure 4. And changes in the text have been made accordingly.
- A discussion has been added in Section 4.3 involving the temperature variability within a single sedimentary facies.
- A discussion has been added in Section 4.3 involving the limited possibility that our SST reconstruction is influenced by the migration of GDGT export within the water column.

[revised manuscript text omitted]